# Robust Isochron Calculation

Roger Powell[1], Eleanor CR Green[1], Estephany Marillo Sialer[1], and Jon Woodhead[1]

[1]School Earth Sciences, The University of Melbourne, Vic 3010, Australia

**Correspondence:** Roger Powell (powell@unimelb.edu.au)

**Abstract.**

The standard classical statistics approach to isochron calculation assumes that the distribution of uncertainties on the data arising from isotopic analysis is strictly Gaussian. This effectively excludes from consideration datasets that have more scatter, even though many appear to have age significance. A new approach to isochron calculations is developed in order to circumvent this problem requiring only that the central part of the uncertainty distribution of the data defines a "spine" in the trend of the data. This central spine can be Gaussian but this is not a requirement. This approach significantly increases the range of datasets from which age information can be extracted but also provides seamless integration with well-behaved datasets, and thus all legacy age determinations. The approach is built on the robust statistics of Huber (1981), but using the data uncertainties for the scale of data scatter around the spine, rather than a scale derived from the scatter itself, ignoring the data uncertainties. This robust data-fitting reliably determines the position of the spine when applied to data with outliers, but converges on the classical statistics approach for datasets without outliers. The spine width is determined by a robust measure, the normalised median absolute deviation of the distances of the data points to the centre of the spine, divided by the uncertainties on the distances. A test is provided to ascertain that there is a spine in the data, requiring that the spine width is consistent with the uncertainties expected for Gaussian-distributed data. An iteratively-reweighted least squares algorithm is presented to calculate the position of the robust line and its uncertainty, accompanied by an implementation in Python.

## 1 Introduction

The ability to fit a straight line through a body of isotope ratio data in order to form an isochron is the cornerstone of many geochronological methods. In detail, however, this is a non-trivial task, since uncertainties are usually associated with all variables, and these are often correlated, precluding simple "least squares" line-fitting techniques. Most of the research in this area was conducted in the late 1960's and early 1970's, being dominated by a classical statistics approach in which data uncertainties, derived from the analytical methods, are taken to be strictly Gaussian-distributed (e.g. York, 1969; York et al., 2004, and references therein). This approach, referred to here as YORK, became entrenched in the geochemical community, particularly in the last two decades as the essential component of the very widely-used software, ISOPLOT, e.g. Ludwig (2012).

In this contribution we examine some of the problems inherent in these techniques and suggest an alternative approach. Our primary focus here will be on general-purpose isochron calculations that determine the age of an "event" that established the isotopic compositions of samples in a dataset. This involves what are called model 1 and 2 calculations in ISOPLOT - as

described below. Approaches that try to extract detail within events, including ISOPLOT model 3 calculations, are not considered (but see e.g. Vermeesch, 2018).

## 1.1 On ISOPLOT

In order to show that there are significant problems in using ISOPLOT for general-purpose isochron calculations, and then to see how they can be addressed, it is first necessary to outline the ISOPLOT protocol, some details of which may not be apparent to the end user. Central to the ISOPLOT workflow, the main tool for considering data scatter is `mswd`, the mean standard weighted deviates (also called the reduced chi-squared statistic), see eq. 1. For strictly Gaussian distributed uncertainties $(n-2)$`mswd` is distributed as chi-squared ($\chi^2_{n-2}$), meaning that if uncertainties are correctly assigned, a strong statistical statement can be made

about whether the scatter of a particular dataset is solely consistent with the associated data uncertainties (i.e. with no geological scatter), for example in the form of a 95% confidence interval on `mswd`, Wendt and Carl (1991). Such a confidence interval is not fixed, but depends on the number of datapoints under consideration, so for example for $n = 10$, `mswd` $< 1.94$ (meaning that `mswd` extends from less than 1 through to 1.94), while for $n = 50$, `mswd` $< 1.36$. A dataset with `mswd` in the chosen range for the number of datapoints has data scatter that is consistent with the data uncertainties. This situation is commonly referred to as

`mswd` "passes"; otherwise `mswd` "fails", in relation to $\chi^2_{n-2}$. The situation where `mswd` passes provides a "pure" interpretation of YORK, and, in ISOPLOT is referred to as a model 1 isochron fit. This is depicted as the horizontal line in Fig. 1, indicating that in this range of `mswd`, corresponding to a confidence interval, the calculated uncertainty on an isochron age does not vary with `mswd`. Such a figure is drawn by taking an actual dataset and progressively modifying it to show what happens as `mswd` varies, as described in Appendix A.

What if `mswd` is greater than the upper limit of the chosen confidence interval? Then the data are considered to have *excess* scatter, in addition to that accounted for by the data uncertainties (assuming that they are strictly Gaussian)[1]. At this point, ISOPLOT, asks the user whether an alternative—model 2—calculation should be undertaken. This decision point is indicated at A in Fig. 1. If the user declines, ISOPLOT gives results that are referred to here as model 1x, as shown in Fig. 1. The model 1 age uncertainty is multiplied by $\sqrt{\text{mswd}}$, to reflect the data scatter being more than expected from the data uncertainties alone.

With further scatter, the switch is made to model 2, at B in Fig. 1. Alternatively, if the user accepts the switch to model 2, then model 1x is not used, indicated by the vertical line at A extending up to the model 2 line in Fig. 1. The model 2 calculation in ISOPLOT is unrelated to YORK. The data uncertainties are discarded and the slope of the line through the linear trend is calculated as the geometric mean of the lines calculated by unweighted least squares of $y$ on $x$ and of $x$ on $y$ (see Appendix A).

In summary, then, in ISOPLOT the calculation of ages and their uncertainties involves a number of decision points based around the concept of `mswd` that impart significant (and in our view unwelcome) step-changes in the way that the data are handled, and algorithms applied. To assist in further discussion of these matters we depart from the language of ISOPLOT at this point, reintroducing the term errorchron, counterposed to isochron, following Brooks et al. (1972). The idea is that

---

[1]excess scatter occurs when the data distribution has higher variability in the tails than is indicated by the variability of the central part of the distribution, for example if the distribution is Gaussian in the centre but having "fat" tails

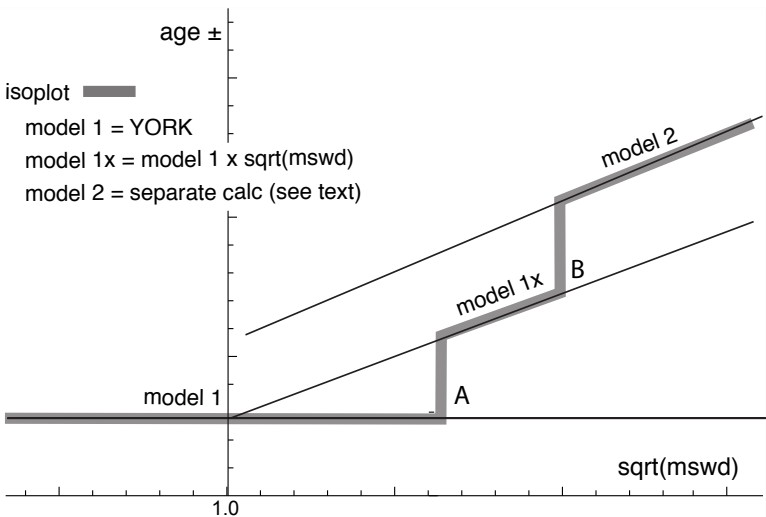

**Figure 1.** Age uncertainty (age$\pm$) plotted against $\sqrt{\texttt{mswd}}$ under the ISOPLOT protocol for a progressively modified dataset (see text, and Appendix A). Under the condition of a model 1 fit, the age uncertainty is constant with increasing data scatter (reflected in increasing $\texttt{mswd}$), until there is a step change in the data treatment at A when the age uncertainty is multiplied by sqrt(mswd). Then at B there is another step change in age uncertainty calculation with increasing data scatter forming ISOPLOT model 2 (see text)

isochrons have a higher chance of having age significance, while errorchrons have a lower chance. In particular, it seems to
be unhelpful for the results of model 2 calculations to be called isochrons as is the case in ISOPLOT, given that there is excess scatter in the data.

## 1.2  Replacing ISOPLOT

Given that ISOPLOT's implementation of model 1+2 line fits is the gold standard of isochron calculations presently, where are the problems, and then what can be done to address them? A shortcoming in YORK stems from the assumption that data
uncertainties are *strictly* Gaussian-distributed. In real-world application this appears to be too restrictive, with datasets that are likely to have age significance being labelled as errorchrons because $\texttt{mswd}$ is too large. While using YORK guided by $\texttt{mswd}$ is optimal statistically if data uncertainties are strictly Gaussian, this logic fails once uncertainties are even slightly non-Gaussian (originally Tukey, 1960). In such circumstances, both $\texttt{mswd}$ and least squares methods themselves, like YORK, become unreliable (e.g. Hampel et al., 1986; Huber, 1981).
Rather than being truly Gaussian, data uncertainties may well be Gaussian-distributed in their centres, but slightly fat-tailed distant from the centres. An isotopic dataset looks intuitively acceptable if the data has a central linear "spine", in which scatter is commensurate with stated analytical uncertainty, but this spine is flanked by data of somewhat larger scatter (i.e. excess scatter, from the "fat tail"). This excess scatter may originate in the isotopic analysis or as a result of geological disturbance.

Age-significance in such data manifests primarily via the position of the spine. In the following, we focus on this spine in the data.

Adopting this spine approach, a successful calculation method for a dataset that may not have strictly Gaussian-distributed uncertainties must, firstly, ascertain whether or not such a spine exists in the data—and hence whether calculations yield an isochron or an errorchron. Secondly, in the case of an isochron calculation, the successful method must reliably locate the spine without being perturbed by vagaries in the more scattered data. Classical statistical methods can do neither of these things, tending to be excessively influenced by the data at the extremes of the scatter. However, the field of robust statistics offers calculation methods that can. When a dataset has no excess scatter, reflected in mswd lying within an appropriate $\chi^2$-constrained confidence interval, such methods can be devised to retrieve identical (or nearly identical) results to classical statistic methods, but, in addition, provide reliable age and age-uncertainty estimates in the presence of excess scatter around a spine. This continuity of operation with increasing mswd contrasts with previous approaches and means that the steps in the ISOPLOT line in Fig. 1, which are certainly undesirable, are circumvented. Moreover the involvement of potentially unreliable least squares-based methods, like ISOPLOT model 2, are avoided when the data show excess scatter.

## 2   An Algorithm for Isochron Calculations

An algorithm is sought that finds a robust straight line through a 2-dimensional linear data trend, while converging with the classical statistical approach of YORK for datasets with consistent scatter (i.e. mswd passes). This section describes the nature of the problem and the theoretical basis for the robust statistical approach that will be adopted. The algorithm adopted

1. determines a preliminary fit of the data, not dependent on vagaries of the data scatter

2. determines the spine width in relation to this preliminary fit

    – if the spine width is in an acceptable range: isochron

    – if the spine width is not in an acceptable range: errorochron

3. determines a robust fit of the data, starting from the preliminary fit, this fit converging with YORK for "good" data

This algorithm is fleshed out below, and then evaluated via simulated datasets and applied to a natural dataset. The central calculation in the algorithm is detailed in Appendix B, and a `python` implementation is provided in Appendix C.

### 2.1   Uncertainty distributions and data fitting

Geochronological datasets are collected on the presumption that the isotopic compositions were established via an "event" the age of which is to be estimated. Given the focus here on data with linear trends, even if the effect of the event is recorded perfectly by the samples analysed—the isotopic compositions lying on a line—the actual data are measured with finite precision and so the data inevitably scatter about the trend. An uncertainty probability distribution can be used to describe the form of the data scatter.

Classical statistical methods assume that the underlying uncertainty distribution of a dataset is known, typically taken to be Gaussian. Under the Gaussian assumption, if the analytical uncertainty on the measurements have been appropriately inferred, `mswd`, the classical statistics parameter used in YORK to validate an isochron, tests that the scatter of datapoints is consistent with the inferred uncertainties. But, in general, there is no reason to suppose that a given analytical technique generates a truly Gaussian uncertainty distribution. Even small amounts of geological disturbance destroy the optimality of YORK. If the uncertainty distribution is not strictly Gaussian then classical methods of data fitting become sub-optimal or worse.

While there are many possible non-Gaussian uncertainty distributions, this paper is concerned with a situation commonly occurring in datasets, in which the datapoints form a linear spine with Gaussian-like scatter, but additional scatter is seen in the tails of the distribution. Such a dataset still encodes meaningful age information in its spine, yet it will typically fail an `mswd` test owing to its departure from a Gaussian distribution. In this work, datasets of this nature are modelled using a contaminated Gaussian uncertainty distribution (Gaussian mixture) (e.g. Tukey, 1960). Such distributions are written $c\%d$N, meaning that with a probability $(100 - c)\%$ the distribution involves a standard deviation, $\sigma$, but with a probability $c\%$ the distribution has a standard deviation, $d\sigma$, both with a mean of zero (see Powell et al., 2002; Maronna et al., 2019, Sect. 2.1). An example is 25%3N, with $c = 25$ and $d = 3$, so that with 25% probability the uncertainty is drawn from $\mathrm{N}(0, 3\sigma)$, and 75% probability drawn from $\mathrm{N}(0, \sigma)$, with the $\mathrm{N}(0, s)$ notation indicating a Gaussian distribution with a mean of zero and a standard deviation of $s$. Such distributions provide excess scatter suitable for developing and evaluating a robust line-fitting calculation. It does not matter if excess scatter in real data is drawn from a different distribution. Note that with the sample sizes provided by most modern geochronological techniques, it is not necessarily possible to test for Gaussian behaviour, or such departures from Gaussian behaviour, although they may be evident on quantile-quantile plots (see below).

## 2.2 Isochrons and errorchrons

In YORK, assuming that the data uncertainties are strictly Gaussian distributed, the probability distribution of `mswd` provides bounds that can be used to distinguish isochrons from errorchrons (e.g. Wendt and Carl, 1991). These bounds come from a 95% confidence interval on `mswd`, as discussed in Appendix A. Datasets whose scatter give `mswd` outside the bounds are deemed to be errorchrons, not isochrons. The focus in this paper is on `mswd` that is too large, indicating excess scatter. Mswd is defined with the residuals, $r_k$, the distance in $y$ of the datapoint, $k$, to the line, $e_k$, weighted by the uncertainty on this distance, $\sigma_{e_k}$

$$\texttt{mswd} = \frac{1}{n-2} \sum_{k=1}^{n} r_k^2 \qquad \text{with} \qquad r_k = \frac{e_k}{\sigma_{e_k}} = \frac{a + bx_k - y_k}{b^2\sigma_{x_k}^2 + \sigma_{y_k}^2 - 2b\sigma_{x_k}\sigma_{y_k}\rho_{x_ky_k}} \tag{1}$$

The line being fitted is $y = a + bx$; datapoint, $k$, is $\{x_k, y_k\}$; the analytical uncertainty on $x_k$, $\sigma_{x_k}$; the analytical uncertainty on $y_k$, $\sigma_{y_k}$; and the correlation between $x_k$ and $y_k$, $\rho_{x_ky_k}$ (see derivation of eq. B4 in Appendix B). Note that the slope, $b$, appears in the denominator of $r_k$, as well as the numerator.

If, instead, data uncertainties are $c\%d$N, with unknown $c$ and $d$, or some other contaminated Gaussian distribution then there is no equivalent of the `mswd` argument to say which datasets should give isochrons rather than errorchrons. The approach advocated here is to use a measure that reflects whether the dataset has a linear spine of "good" data within it. The measure

suggested, $s$, coined the spine width, is robust, and is defined as

$$s = \texttt{nmad}(r) = 1.4826 \, \texttt{median}(|r_k - \texttt{median}(r)|) \tag{2}$$

with a normalisation constant[2], 1.4826. Given that $s$ is based on a median, its magnitude depends on that half of the data that have the smallest absolute values of centred $r$, in other words those that would define a spine. If the data were in fact Gaussian-distributed, it is expected that $s$ should be in a range about 1 in the same way that $\texttt{mswd}$ is, given that $r$ already involves the analytical uncertainties. The larger is $s$, greater than 1, the less pronounced is the linear spine in the data (or the uncertainties have been underestimated). Whereas the 95% confidence interval (95%ci) on $\texttt{mswd}$ for Gaussian-distributed uncertainties comes from a well-established probability distribution, with $(n-2)\texttt{mswd} \sim \chi^2_{n-2}$ (e.g Wendt and Carl, 1991), the confidence interval on $s$ needs to be found by simulation (see Appendix D), with the simulated datasets just involving Gaussian-distributed uncertainties. The intervals are given in this table, Table 1:

| | 95%ci | $\sqrt{\texttt{mswd}}$ | | 95%ci | $s$ | |
|---|---|---|---|---|---|---|
| $n$ | low | high | $*$ | low | high | $*$ |
| 5 | 0.268 | 1.765 | 1.614 | 0.09 | 1.64 | 1.48 |
| 6 | 0.348 | 1.669 | 1.540 | 0.17 | 1.62 | 1.47 |
| 8 | 0.454 | 1.552 | 1.449 | 0.26 | 1.58 | 1.45 |
| 10 | 0.522 | 1.480 | 1.392 | 0.31 | 1.55 | 1.43 |
| 15 | 0.621 | 1.379 | 1.312 | 0.40 | 1.50 | 1.40 |
| 30 | 0.739 | 1.260 | 1.215 | 0.58 | 1.39 | 1.33 |
| 60 | 0.818 | 1.181 | 1.151 | 0.71 | 1.28 | 1.23 |

Whereas 1-sided confidence intervals are advocated in Appendix A—columns marked with an asterix in the Table—2-sided confidence intervals are also given in the table. Using the asterixed column, for example for a dataset with 10 datapoints ($n = 10$), the dataset is deemed to yield an isochron if the observed $s$ is less than 1.43. If $s$ is larger, the dataset gives an errorchron. For isochrons, the age uncertainty is calculated as in Appendix B. For errorchrons, the age uncertainty is not calculated because the data scatter is not consistent with the analytical uncertainties at 95% confidence, $s$ being larger than the bound.

## 2.3 A robust statistics approach to isochron calculation

We seek a statistical approach to isochron calculation that is *robust* (e.g. Huber, 1981; Hampel et al., 1986), meaning that it is not excessively affected by outliers in the data, while having desirable statistical properties, for example good efficiency (see below). In addition, we require the approach to converge to YORK for a "good" dataset, one with a near-Gaussian uncertainty distribution, allowing seamless compatibility with classical data interpretation. The overall approach adopted will be referred

---

[2]the normalisation constant makes $\texttt{nmad}(r)$ an unbiased estimator of the standard deviation when the data are Gaussian distributed as the sample size becomes large (e.g. Maronna et al., 2019, Sect. 2.4)

to as SPINE, involving the use of spine width for isochron-errorchron distinction, combined with robust line-fitting. The line-fitting is based on the approach of Huber (1981), as outlined in Maronna et al. (2019), Sect. 2.2.2. Whereas most robust line-fitting methods use the scatter of the data as a scale, data uncertainties having been discarded (e.g. Powell et al., 2002), here the data uncertainties are used. This is necessary in order to preserve continuity of the results with YORK, in which the data uncertainties are an integral part of the calculation.

In both the Huber approach in SPINE, and in YORK, a straight line is fitted to a dataset by minimising a function of the residuals, $r_k$. In the case of YORK this is just the mswd, (1). Since isochron data are generally bivariate with correlated analytical uncertainties in $x$ and $y$, the analytical uncertainty in datapoint $k$ can be represented as an ellipse as in Fig. 2. The absolute value of the residual for datapoint, $k$, $r_k$, is in fact the scaling factor on the size of the ellipse required to expand it or reduce it until it touches the best-fit line (Fig. 2).

The function that is minimised to find the best-fit line can be written $\sum \rho(r_k)$ for both YORK and SPINE. Whereas in YORK, $\rho(r_k) = r_k^2$ for all $r_k$, in SPINE $\rho(r_k) = r_k^2$ near the centre of the uncertainty distribution (as in YORK), but downweights datapoints for which the absolute value of the residual is greater than a cut-off value, $h$. Thus, in SPINE, and in Fig. 3, the Huber $\rho$ is

$$\rho(r_k) = \begin{cases} 2\,h\,r_k - h^2 & r_k < -h \\ r_k^2 & \text{if} \quad -h < r_k < h \\ 2\,h\,r_k - h^2 & r_k > h \end{cases} \tag{3}$$

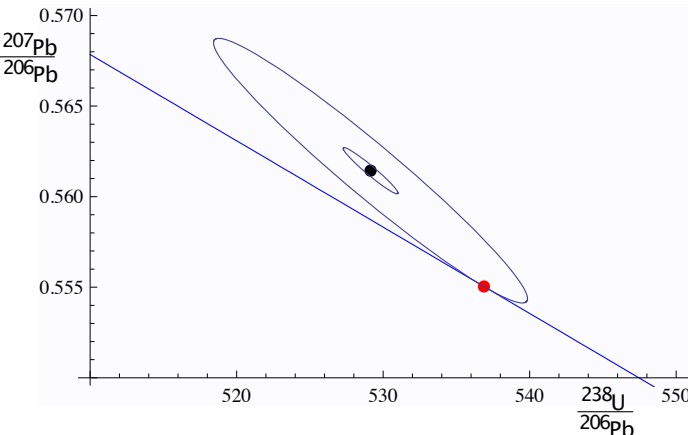

**Figure 2.** For an example datapoint, $\{x_k, y_k\}$, the inner ellipse is calculated with the analytical uncertainties, $V_k$, at the $1\sigma$ level (in black). Given a line, $y = a + bx$ (in blue), the ellipse must be drawn at the $|r_k|\sigma$ level (in red) to touch the line, in this case $|r_k| = 5.73$. The data point is $x_k = 529.14$, $y_k = 0.5614$, and $\sigma_{x_k} = 1.870$, $\sigma_{y_k} = 0.00127$ and $\rho_{x_k y_k} = -0.967$. The line is $y = 0.8108 - 0.0004764\,x$.

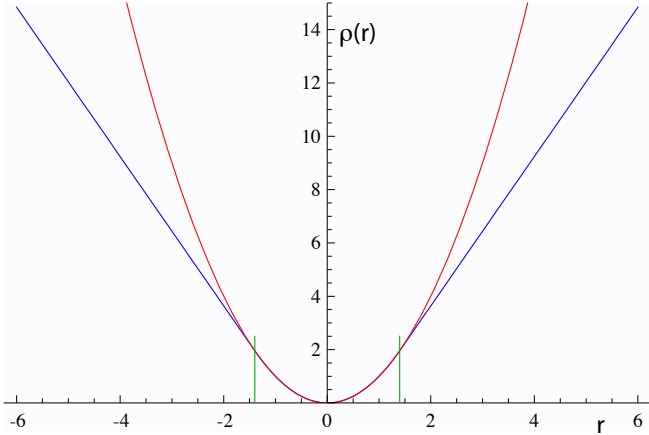

**Figure 3.** Plots of $\rho(r)$ against $r$ for YORK in red ($r^2$), and for Huber in SPINE (eq. 3) in blue. The two curves are coincident for $|r| < h$, with $h = 1.4$ the vertical green lines. See text.

In SPINE, for residuals that have an absolute value less than an adjustable constant, $h$, the contribution to the sum being minimised is the same as for YORK, but it is linear in the residual for larger absolute value. Note that as $h$ becomes larger and larger, SPINE converges to YORK. The value to use for $h$ is discussed in Maronna et al. (2019)[3].

The iteration developed in Appendix B minimises $\sum_k \rho(r_k)$ with respect to the unknown, $\theta$, a two-element column vector, $\{a, b\}^{\mathrm{T}}$ in the line equation, $y = a + bx$. The iteration is applicable to SPINE and also YORK. As a starting point of the iteration, data are fit for $\theta$ with SIEGEL (Siegel, 1982), which is highly robust toward excess scatter. However SIEGEL is much less efficient than SPINE (see below), so SPINE is a better ultimate estimator. A full iteration is envisaged in Appendix B. Once $\theta$ is calculated, the measure of scatter used to distinguish an isochron from an errorchron can be calculated using Table 1. If an isochron is deemed to have been calculated, the uncertainty on $\theta$, $\mathbf{V}_\theta$, can be found, as outlined in Appendix B.

The SPINE algorithm can be summarised:

1. determine a preliminary fit of the data using SIEGEL

2. determine the spine width using `nmad`

    – if the spine width is in an acceptable range, from col. 6 in Table 1: isochron

    – if the spine width is not in the acceptable range: errorchron

3. determine a robust fit of the data, minimising $\sum \rho(r_k)$, with the Huber $\rho(r_k)$, (3), starting from the SIEGEL estimate of $\theta$

---

[3]The value used here, $h = 1.4$, involves a particular tradeoff between robustness and efficiency (Maronna et al., 2019, Sect. 2.2.2 and 3.4)

## 2.4 Application of SPINE to simulated datasets

Assessing algorithms for data fitting is best done using simulated datasets, so that the true "age" represented by the data is known. In this case, datasets were generated by drawing data points from a range of uncertainty distributions, all centred on a linear trend reflecting an age of 4 Ma. Full details are provided in Appendix D. Two features of the datasets are varied: the number of datapoints in the dataset, and the uncertainty structure adopted, the latter via varying $c$ and $d$ in $c\%d$N. The algorithm is assessed in terms of its ability to retrieve the specified age of the linear trend on which the simulated datasets are built, and on the uncertainty in the age.

Given that the datasets investigated have fat-tailed contaminated-Gaussian uncertainty distributions, the focus is on the effect of excess scatter in the data, in other words, data scatter over and above what is expected for Gaussian data uncertainties. Nevertheless a small proportion of datasets do have small scatter, giving $s$ which is below the lower bound for that number of datapoints.

The analysis below compares the results of YORK, applied only to those simulated datasets that lie within the `mswd` bounds, with the results of SPINE applied to those datasets that lie within the spine width ($s$) bounds. The greatest majority of the former are included in the latter, e.g. $> 97\%$ for $n = 10$). Importantly, however, SPINE typically identifies reliable age information in many more datasets than YORK. In the following table, $m\%$excl and $s\%$excl are the percentage of simulated datasets excluded on the basis of the `mswd` and $s$ bounds, respectively:

| $n$ | N | | 5%3N | | 25%3N | | 10%10N | |
|---|---|---|---|---|---|---|---|---|
| | $m\%$excl | $s\%$excl | $m\%$excl | $s\%$excl | $m\%$excl | $s\%$excl | $m\%$excl | $s\%$excl |
| 5 | 2.5 | 2.5 | 8.7 | 4.0 | 30.2 | 9.8 | 32.5 | 9.5 |
| 6 | 2.5 | 2.5 | 9.6 | 3.9 | 34.6 | 13.8 | 37.3 | 10.9 |
| 8 | 2.5 | 2.5 | 12.7 | 4.2 | 44.7 | 14.5 | 46.0 | 10.4 |
| 10 | 2.5 | 2.5 | 14.2 | 4.0 | 51.8 | 15.2 | 53.5 | 9.7 |
| 15 | 2.5 | 2.5 | 17.4 | 4.2 | 65.2 | 17.1 | 68.2 | 9.1 |

Note that, for example, for $n = 10$, datasets drawn from 5%3N, in fact have $100(10^{0.95}) = 59.9\%$ of the datasets having all uncertainties Gaussian, while 40.1% have at least one uncertainty drawn from 3 times Gaussian (3N). For 25%3N, 5.6% are Gaussian only, and for 10%10N, 34.9%. The leftmost columns are 2.5% by definition.

A 95% confidence interval on age can be found by ordering the list of ages calculated for the datasets, selecting the the lower limit at the 2.5% point in the list, and the upper limit at the 97.5% point. For datasets that lie within the $s$ bound in SPINE or `mswd` bound in YORK, the 95% confidence intervals on ages are essentially the same, even though a much larger proportion of datasets lie within bounds using SPINE than using YORK. A comparison of ages can also be made between YORK and SPINE for simulated datasets that lie *outside* the `mswd` bounds. For $n = 10$ and Gaussian-distributed data, the 95% confidence interval is $\pm 0.021$ Ma for both YORK and SPINE, but for 5%3N, 25%3N and 10%10N, the 95% confidence interval on the YORK ages are $\pm 0.035$, $\pm 0.040$, and $\pm 0.092$ Ma, respectively, while the SPINE ages are $\pm 0.027$, $\pm 0.034$, and $\pm 0.034$ Ma. The differences between the ages given by the two methods, normalised by the uncertainties on the SPINE ages, are $\pm 1.62$, $\pm 1.63$ and $\pm 5.54$.

Comparison of the intervals shows that the `mswd` bounds are relevant to the application of YORK, in that generally it does not work well outside the bounds, while indicating that SPINE is (much) more reliable for age determination for such datasets.

Considering age uncertainty, it might be expected that the uncertainties suffer from the excess scatter in the data in datasets that yield an errorchron with YORK but an isochron with SPINE.This appears not to be the case, but there is a small degradation in the age uncertainties retrieved caused by an unavoidable efficiency loss using SPINE. Efficiency at the Gaussian distribution is the ratio of the variance obtained by the optimal estimator (YORK), divided by the variance using the chosen robust estimator (in this case, SPINE). SPINE involves an efficiency loss at the Gaussian distribution, which is illustrated in Figure 4 via kernel density estimate (`kde`) plots of the age uncertainties calculated for simulated datasets with $n = 10$ and Gaussian-distributed uncertainties. Kde plots are probability distributions akin to smoothed histograms (Wand and Jones, 1995). The red curve is the `kde` for datasets that have all $|r_k| < h$, as would be given by YORK on datasets with Gaussian-distributed uncertainties. The blue curve is the `kde` for all datasets with at least one $|r_k| > h$. The overall `kde`, in black, is the `kde` of all of the datasets in the red and blue `kde`, in observed proportion, about 30% to 70%. The efficiency loss of SPINE is reflected in the displacement of the black curve to slightly higher age uncertainty than the red curve. The relationships shown in Figure 4 for $n = 10$ can be seen for other $n$ in Figure 5. The pairs of red and black lines, correspond to, and have the same meaning as the red and black lines in Figure 4. As expected, the distribution of age uncertainties moves towards larger values as the sample size $n$ decreases.

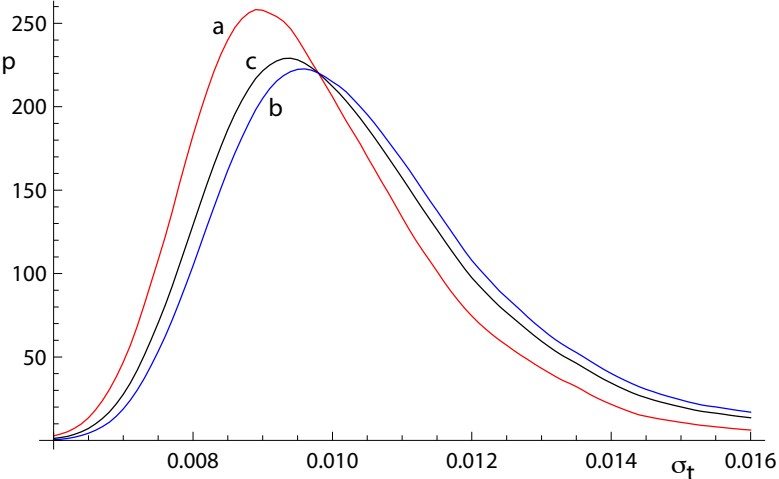

**Figure 4.** Kernel density estimates (`kde`) for age uncertainty calculated with SPINE on 10,000 simulated datasets with $n$=10 and Gaussian-distributed uncertainties. (a) Those datasets for which all $|r_k|<h$ (in red); (b) those datasets for which at least one $|r_k|>h$ (in blue), and (c) overall result combining a and b in observed proportion (in black).

Whereas SPINE involves an efficiency loss at the Gaussian distribution, it performs better—i.e. it is much more robust—than YORK for data from contaminated Gaussian distributions (e.g. Maronna et al., 2019, Table 2.2). The ability of SPINE to retrieve age uncertainty information for such data varies with the probability and scale of the contamination, as shown in Figure 6. Not unexpectedly the more seriously contaminated distributions (25%3N and 10%10N) involve a greater displacement of the `kde`

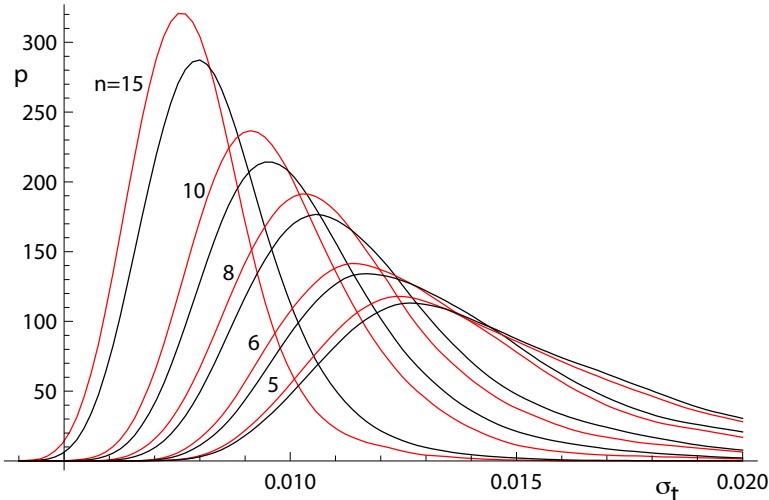

**Figure 5.** Kernel density estimates for age uncertainty calculated with SPINE on 10,000 simulated datasets with a range of $n$ values and Gaussian-distributed uncertainties. In each case, the kde for those datasets for which all $|r_k|<h$ is in red and the overall kde is in black.

to higher age uncertainty than the more weakly contaminated 5%3N distribution. Although the displacement of the blue curves from the black curve is real, nevertheless the ability of SPINE to retrieve age uncertainties from datasets with contaminated distributions is good.

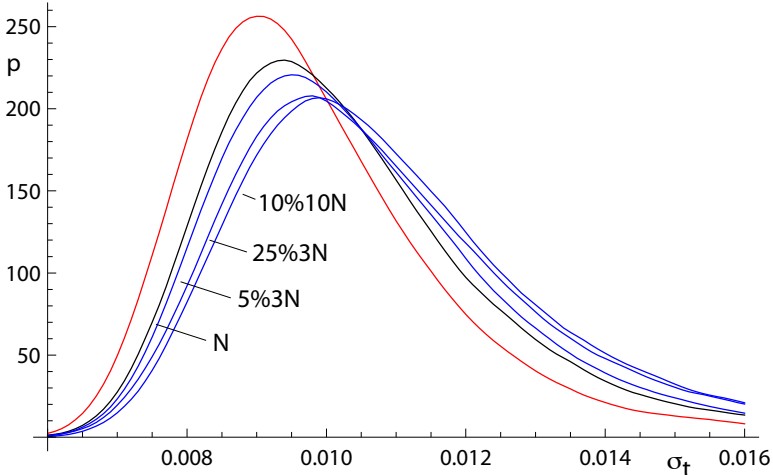

**Figure 6.** Kernel density estimates for age uncertainty calculated with SPINE on 10,000 simulated datasets with $n=10$ and several uncertainty structures. The kde for those datasets for which all $|r_k|<h$ is in red, the kde for datasets with Gaussian-distributed uncertainties is in black, and the kde for all of the datasets for 5%3N, 25%3N and 10%10N respectively are in blue.

## 2.5 Application of SPINE to a natural dataset

In order to show the real-world utility of SPINE, we consider data for a carbonate flowstone from the Riversleigh World Heritage fossil site in Queensland, Australia (Sample 0708). Isotope dilution U-Pb data for the bulk sample were previously published by Woodhead et al. (2016) providing a Model 2 isochron with an age of $13.72 \pm 0.12$ Ma and with a `mswd` of 3.7. The new data presented here were obtained by laser ablation ICPMS on the same sample using methods outlined and published in Woodhead and Petrus (2019). Such datasets are typically larger with little error correlation (rounder error ellipses), but with larger uncertainties than isotope dilution data. These new data define an errorchron under the YORK assumptions, with `mswd` = 1.68, and a model 2 age of $13.68 \pm 0.31$ Ma. These data might therefore be rejected under the `mswd` criterion despite exhibiting a well-developed linear trend. With SPINE, $s = 1.24$, within the $s$ range for an isochron, the age is $13.69 \pm 0.26$ Ma ($\pm$ is $1.96\sigma$). The data for 0708 are plotted in Fig. 7, with 95% confidence ellipses on the datapoints. Further calculations with this sample, comparing the results of our new algorithm with existing approaches are presented in Appendix A.

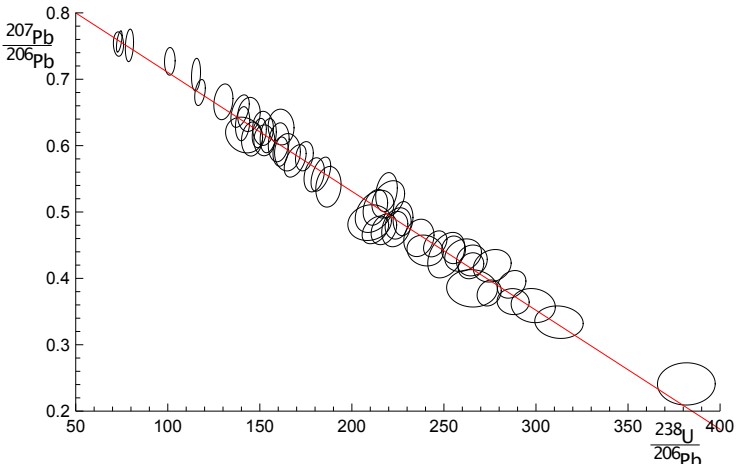

**Figure 7.** Laser ablation example 0708. See text.

With 0708, the SPINE and ISOPLOT ages are very similar, but `mswd` suggests an errorchron, so there is excess scatter on this basis. Quantile-quantile plots can be used to show the nature of the distribution of the residuals that contribute to such excess scatter. Here, Fig. 8, the ordered residuals normalised to $\mathrm{nmad}(r)$ are plotted against the quantiles of the standard Gaussian distribution, $N(0,1)$ (see Fox, 2016, Sect 3.1.3). The normalisation means that if the residuals are Gaussian-distributed, they should plot close to a line of unit slope through the origin. The figure also includes 95% pointwise confidence intervals (blue dashed lines). The figure suggests that the residuals are in fact effectively Gaussian-distributed, lying between the blue-dashed lines (and `mswd` has been too sensitive in flagging that the residuals are not Gaussian-distributed).

Generally, geochronological datasets do not have obvious outliers—they might be "cleaned" of isolated datapoints before an age calculation, or the dataset might even be discarded. But many of the simulated datasets from contaminated Gaussian

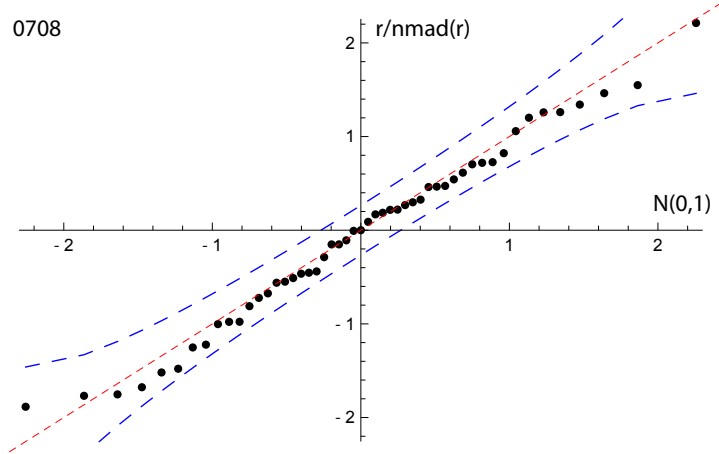

**Figure 8.** Quantile-quantile plot for sample 0708, using the SIEGEL fit of the data to calculate the residuals to plot. The dashed red line has unit slope through the origin; the blue dashed lines are the 95% pointwise confidence intervals. See text.

distributions, as used above, do contain outliers. In Appendix E an example of a simulated dataset with outliers (from 25%3N) is used to show how YORK and SPINE behave, and to show what a quantile-quantile plot looks like in those circumstances.

## 3   Discussion

This work was motivated by the belief that many isotopic datasets contain meaningful age information that cannot be identified using classical statistical methods and may therefore be discarded or discounted. In such cases, the age information is contained in a linear spine in the data, but the data also contains scatter that is inconsistent with a Gaussian uncertainty distribution, having fatter tails than Gaussian. A statistical test based on the spine width is devised, akin to using `mswd` in classical methods, allowing an isochron-errorchron distinction to be made. Many datasets that give isochrons based on the spine width, give errorchrons under the assumption of a strict Gaussian uncertainty distribution. A statistically-robust isochron calculation method is able to retrieve this age information and to provide appropriate uncertainty estimates. Calculated ages and age uncertainties are more reliable than ISOPLOT ones for data with excess scatter.

Contaminated Gaussian distributions provide a model for a type of dataset with excess scatter relative to a strictly Gaussian-distributed one. The robust isochron method presented in this work can however be applied *in general* to data which is Gaussian-distributed only in the central spine of the uncertainty distribution, with non-Gaussian scatter occurring in the tails, arising from analytical or geological uncertainty. Indeed, it can be applied to any dataset with a well-developed spine in the data.

In most robust statistics data fitting approaches, the formal uncertainties output during data measurement are ignored. Instead, the scale used in the data fitting is derived from the scatter in the data themselves, an approach adopted by Powell et al. (2002). The new approach followed here *does* include the data measurement uncertainties, and this allows the results to converge on those of YORK when the data lack excess scatter. This provides compatibility with older "good" datasets processed

using the classical statistical approach but, going forward, allows age information to be extracted from a much wider range of datasets which might otherwise be rejected for having `mswd` greater than the isochron cutoff.

A problem with SPINE, shared with YORK, is that the effect of high-leverage data is not taken into account. Such data are easily recognised in $x$-$y$ plots, when a small proportion of the data—even one datapoint—is separated from the main body of the data along the trend through the data. Data fitting tends to be overly constrained to fit high-leverage data, giving them small residuals, even if the best fit of the main body of the data alone would give the high-leverage data larger residuals. In 0708, the point at highest $x$ is relatively high leverage (`hat` $= 0.171$). Robust approaches have been developed to handle high leverage data, e.g. Maronna et al. (2019), ch. 5, but are not yet developed for the situation where the data uncertainties are taken into account, nor for the relatively small datasets that are typical of geochronological studies (c.f. Fig. 7). Huber and Rochetti (2009), ch. 7, have a counter view advocating data assessment, rather than aiming for a black-box method to try and automatically safeguard against the potentially deleterious effect of high leverage data, an approach we suggest here. In the case of the relatively high leverage datapoint in 0708, omitting this datapoint gives $13.75 \pm 0.27$ Ma (compare row 11 with row 1 in the Table in Appendix A), within uncertainty of the age including this datapoint.

**Appendix A: Algorithms and applications to sample 0708**

Here are collected some results of calculations for sample 0708, used in Fig. 7, and some related algorithmic details.

**thought experiment in Fig. 1**

The thought experiment sketched in Fig. 1 aimed to show the consequence for ISOPLOT behaviour of the modification of the observed data in a dataset to reduce or increase the scatter about the linear trend. The calculated equivalent of Fig. 1 for sample
0708 is shown in Fig. A1, including also the corresponding SPINE results. In calculating the Figure, the modification of the dataset is achieved by first taking the data points with their attendant error ellipses (i.e. covariance matrices), and moving them all in to lie on the linear trend, considered as fixed by a YORK calculation. Then the points and ellipses are considered to be displaced away from the trend. This is "move" in the Table below, going from -1, when the points lie on the trend, through 0, with the points as in the original data, to positive when displaced further away from the trend. "Move" varies more or less
linearly with $\sqrt{\mathrm{mswd}}$ from -0.133 at the left-edge of the Figure, to 0.064 on the right-edge; the spine width changes from 1.08 to 1.32 across the Figure. For these calculations the last line in the dataset is omitted as it is relatively high leverage (hat = 0.171), not wishing this datapoint to affect the results.

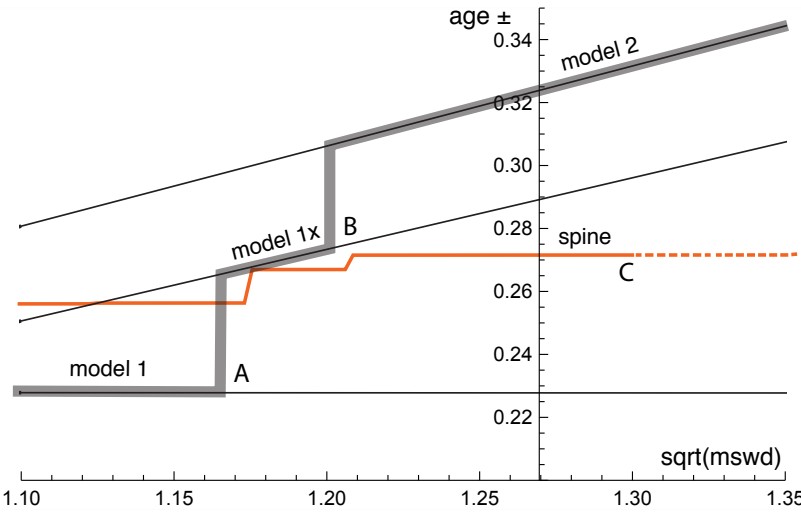

**Figure A1.** Age uncertainty (age±) plotted against $\sqrt{\mathrm{mswd}}$ under the ISOPLOT protocol for the progressively modified dataset, 0708 (see text). In model 1, the age uncertainty is constant with increasing data scatter (reflected in increasing $\mathrm{mswd}$), until there is a step change in age uncertainty at A when the ± is multiplied by $\sqrt{\mathrm{mswd}}$. Then at B there is another step change with further increase of data scatter to model 2 (see text). The location of the step at A is based on a 95% confidence interval for $\mathrm{mswd}$ (discussed below), whereas the $\sqrt{\mathrm{mswd}}$ position of the step at B is arbitrary. The $y$-axis is drawn at the $\sqrt{\mathrm{mswd}}$ of the actual data, 1.27 (i.e. no modification of the data). See text.

   In the Figure, extending from the left, through $\mathrm{mswd} = 1$, to A, the ISOPLOT age uncertainty (model 1, i.e. YORK) is constant because the data scatter is consistent with the data uncertainties. Through this range the SPINE age uncertainty is above the

305 model 1 line because of the efficiency loss embodied in SPINE, as shown in Figs 4–5. However, after the age uncertainty steps with increasing `mswd`, to the right of the diagram, the SPINE age uncertainty is smaller than the model 1x and model 2 age uncertainty. This is because SPINE gives an isochron on the basis of spine width up to C at $\sqrt{\texttt{mswd}} \approx 1.3$, whereas model 2 is an errorchron, on the basis of the assumption of strictly Gaussian data uncertainties. The small steps in the SPINE age uncertainty line are an artefact of the approximation used in the calculation of the age uncertainty, see Appendix B.

In the top part of the following Table, the results for SPINE and for ISOPLOT are summarised. The $\Delta$ column gives the change to the age from the SPINE age, normalised by the uncertainty on the SPINE age. Below the double line in the Table, are some results from the above thought experiment, Fig. A1.

|   |   | move | $\sqrt{\texttt{mswd}}$ | age | age$\pm$ | $\Delta$ | notes |
|---|---|---|---|---|---|---|---|
| 1 | SPINE |  |  | 13.685 | 0.257 | − | s = 1.24 |
| 2 | YORK |  | 1.296 | 13.733 | 0.216 | 0.37 | outside 95% c.l. |
| 3 | ISOPLOT model 1x |  |  | 13.733 | 0.280 |  |  |
| 4 | ISOPLOT model 2 |  |  | 13.679 | 0.306 | −0.05 |  |
| 5 | siegel |  |  | 13.803 |  | −0.95 |  |
| 6 | SPINE | −0.133 | 1.1 | 13.769 | 0.252 |  | $s = 1.08$ |
| 7 | YORK |  |  | 13.800 | 0.223 | −0.39 | inside 95% c.l. |
| 8 | SPINE | −0.055 | 1.2 | 13.756 | 0.262 |  | $s = 1.18$ |
| 9 | ISOPLOT model 1x |  |  | 13.800 | 0.268 | −0.39 |  |
| 10 | ISOPLOT model 2 |  |  | 13.832 | 0.300 | −0.62 |  |
| 11 | SPINE | 0 | 1.27 | 13.747 | 0.267 | − | $s = 1.25 \, \Delta = 0.366$ |
| 12 | ISOPLOT model 2 |  |  | 13.836 | 0.317 | −0.65 |  |
| 13 | SPINE | 0.024 | 1.3 | 13.743 | 0.267 |  | $s = 1.26$ |
| 14 | ISOPLOT model 2 |  |  | 13.837 | 0.325 | −0.66 |  |

Additional algorithmic details for ISOPLOT follow next, in part related to the above thought experiment.

**ISOPLOT model 1**

In the ISOPLOT *model 1* calculation, i.e. in YORK, a decision has to be made about the confidence interval on `mswd` that is used to denote the range of data scatter (i.e. `mswd`) that is considered to be accounted for by the data uncertainties, without need to either multiply the age uncertainty by $\sqrt{\texttt{mswd}}$ (i.e. model 1x), or switch directly to an alternative calculation (which is model 2 in ISOPLOT). On the understanding that data uncertainties are correctly assigned, a one-sided confidence interval on `mswd`
can be adopted, acknowledging that `mswd` is not being used to identify the case where assigned data uncertainties are too large or too small. The upper end of the confidence interval on `mswd` is where excess scatter is considered to start, a conventional choice being derived from a 95% confidence interval. Note that there is no argument that this should be at `mswd = 1` (c.f.

Dickin, 2005, p37), unless the number of datapoints is huge. Even for a dataset of 50 datapoints the 95% confidence interval on `mswd` extends to 1.36. In terms of the *probability of fit* measure used in ISOPLOT, this is $100 - 95 = 5\%$. It might be noted that the naming of probability of fit seems unhelpful - it is clearer to focus on `mswd`.

### ISOPLOT model 2

The so-called *error-in-variables* (`eiv`) or measurement-error problem is avoided in YORK because the uncertainty in the $x$ variable is taken into account explicitly. If it was not, then `eiv` results in the calculated slope being biased downwards and the approach being inconsistent (e.g. Fuller, 1987).

In the ISOPLOT *model 2* calculation, `eiv` is avoided, even though the data uncertainties are discarded, by making the slope of the line through the data be the geometric mean of the slopes of ordinary least squares of $y$ on $x$, $b_{yx}$, and $x$ on $y$, $1/b_{xy}$. These are

$$b_{yx} = \frac{\sum (x_k - \overline{x})(y_k - \overline{y})}{\sum (x_k - \overline{x})^2} \qquad \text{and} \qquad \frac{1}{b_{xy}} = \frac{\sum (x_k - \overline{x})(y_k - \overline{y})}{\sum (y_k - \overline{y})^2}$$

with $\overline{x} = \frac{1}{n} \sum x_k$ and $\overline{y} = \frac{1}{n} \sum y_k$. Then

$$b = \pm\sqrt{b_{yx} b_{xy}} = \pm\sqrt{\frac{\sum (y_k - \overline{y})^2}{\sum (x_k - \overline{x})^2}} \qquad \text{and} \qquad a = \overline{y} - b\overline{x}$$

with, in this case, the sign of the square root being negative. The calculation in ISOPLOT does not use this explicit formula, instead adopting an algebraic equivalent that allows the YORK iteration to be used.

### ISOPLOT robust

In ISOPLOT, there is an option to use a *robust* isochron calculation method. The two available have high breakdown point but low efficiency (e.g. Huber, 1981, Sect. 1.2.3). The second method—(Siegel, 1982)—can be considered to supercede the first. In fact, here, SIEGEL is used in the implementation of SPINE as a starting point for the iteration (see Appendix C). The fit of the data for sample 0708 with SIEGEL is given in line 5 of the Table.

### Appendix B: Iteration in SPINE

The SPINE algorithm involves minimising $\sum_k \rho(r_k)$ with respect to the unknown, $\theta$, a two-element column vector, $\{a, b\}^{\mathrm{T}}$ in the line equation, $y = a + bx$, in order to fit the data. The residual, $r_k$ on datapoint $k$ is defined below, and the function $\rho$ is defined in (3) in the main text. The SPINE algorithm subsumes YORK.

Writing the $k$th datapoint as $\{x_k, y_k\}$, generally the isotopic data used in isochron calculations involve uncertainties in both $x_k$ and $y_k$, and commonly the $x_k$ and $y_k$ are also correlated. These can be represented by a covariance matrix, $V_k$,

$$V_k = \begin{bmatrix} \sigma_{x_k}^2 & \sigma_{x_k}\sigma_{y_k}\rho_{x_k y_k} \\ \sigma_{x_k}\sigma_{y_k}\rho_{x_k y_k} & \sigma_{y_k}^2 \end{bmatrix} \tag{B1}$$

in which $\sigma_{x_k}$ is the standard deviation on $x_k$, $\sigma_{y_k}$ the standard deviation on $y_k$, $\sigma_{x_k}\sigma_{y_k}\rho_{x_k y_k}$ the covariance between $x_k$ and $y_k$, and $\rho_{x_k y_k}$ the correlation coefficient between $x_k$ and $y_k$. The covariance matrix can be represented by an ellipse around the data point in an $x$–$y$ diagram, as illustrated in Fig. 2. The residual, $r_k$, a measure of the distance of the point $\{x_k, y_k\}$ to the line, is calculated from the coordinates of the data points, $\{x_k, y_k\}$, and their uncertainties in $V_k$, by

$$r_k = \frac{e_k}{\sigma_{e_k}} \tag{B2}$$

in which $e_k$ is the distance of the datapoint from the line, $e_k = a + b x_k - y_k$, and $\sigma_{e_k}$ is the standard deviation on $e_k$. The standard deviation, $\sigma_{e_k}$, is calculated by error propagation using $V_k$:

$$\sigma_{e_k}^2 = \left\{ \frac{\partial e_k}{\partial x_k}, \frac{\partial e_k}{\partial y_k} \right\} V_k \left\{ \frac{\partial e_k}{\partial x_k}, \frac{\partial e_k}{\partial y_k} \right\}^{\mathrm{T}} = b^2 \sigma_{x_k}^2 + \sigma_{y_k}^2 - 2 b \sigma_{x_k} \sigma_{y_k} \rho_{x_k y_k} \tag{B3}$$

with the term in curly brackets evaluating to $\{b, -1\}$. The residual is then

$$r_k = \frac{e_k}{\sigma_{e_k}} = \frac{a + b x_k - y_k}{\sqrt{b^2 \sigma_{x_k}^2 + \sigma_{y_k}^2 - 2 b \sigma_{x_k} \sigma_{y_k} \rho_{x_k y_k}}} \tag{B4}$$

In matrix form, the residuals can be written as a column vector, $\mathbf{r}$

$$\mathbf{r} = \mathbf{W}_e \mathbf{e} = \mathbf{W}_e (\mathbf{X}\theta - \mathbf{y}) \tag{B5}$$

with $\mathbf{W}_e$ a diagonal matrix with $kk$th element, $1/\sigma_{e_k}$.

The minimisation of $\sum_k \rho(r_k)$ is iterative, starting from a robust but low efficiency estimate of the line, for example using Siegel (1982). The minimisation of $\sum \rho(r_k)$ is undertaken using the fact that, at the minimum, the derivative of $\sum \rho(r_k)$ with respect to $\theta$ is zero. Defining

$$2\psi(r_k) = \frac{\partial \rho(r_k)}{\partial r_k} \tag{B6}$$

this function, for the Huber (1981) approach, from (3), is

$$\psi(r_k) = \begin{cases} -h & r_k < -h \\ r_k & \text{if} \quad -h < r_k < h \\ h & r_k > h \end{cases} \tag{B7}$$

For YORK, $\psi(r_k) = r_k$, equivalent to SPINE with large $h$.

At the minimum of $\sum \rho(r_k)$

$$\sum_k \frac{\partial \rho(r_k)}{\partial \theta} = 0 = \sum_k \left( \frac{\partial \rho(r_k)}{\partial r_k} \right) \left( \frac{\partial r_k}{\partial \theta} \right) = \sum_k \psi(r_k) \left( \frac{\partial r_k}{\partial \theta} \right) = \sum_k \psi(r_k) \frac{1}{\sigma_{e_k}} \left\{ \left( \frac{\partial e_k}{\partial \theta} \right) - r_k \left( \frac{\partial \sigma_{e_k}}{\partial \theta} \right) \right\} \tag{B8}$$

In the curly braces, writing the first derivative, $\{1, x_k\}$, as the $k$th row of a matrix $\mathbf{X}$, and writing the first and second derivatives together as the $k$th row of a matrix $\mathbf{X}'$, then the $k$th row of $\mathbf{X}'$ is

$$X_k' = \{1, x_k'\} = \left\{ 1, x_k - \frac{r_k}{\sigma_{e_k}} \left( b \sigma_{x_k}^2 - \sigma_{x_k} \sigma_{y_k} \rho_{x_k y_k} \right) \right\}$$

with $b$ the slope of the line. The second column of $\mathbf{X}$ is simply the data $x$ values, while the second column of $\mathbf{X}'$ is the $x$ values on the line where the uncertainty ellipses around each datapoint would touch it, $\mathbf{x}'$. In matrix form, (B8) can be written as

$$\mathbf{X}'^{\mathrm{T}}\mathbf{W}_e\psi(\mathbf{r}) = 0 \tag{B9}$$

in which $\psi(\mathbf{r})$ is a column vector whose $k$th element is $\psi(r_k)$. B9 constitutes two non-linear equations requiring iteration to solve. Two iteration schemes are proposed. The first iteration, in fact used for all the simulations, proceeds directly from (B9), while the second iteration is in iteratively reweighted least squares form (e.g. Maronna et al., 2019, Sect. 4.5.2).

In the first scheme, at iteration $i$, progressing towards the minimum with $\theta_i = \theta_{i-1} + \Delta\theta$, for $|r_k| < h$, $\psi_i(r_k) = \psi_{i-1}(r_k) + X_k\,\Delta\theta/\sigma_{e_k}$ and $\psi_i(r_k) = \psi_{i-1}(r_k)$ otherwise. This can be written $\psi_i(r_k) = \psi_{i-1}(r_k) + \dot{\psi}_{i-1}(r_k)\,X_k\,\Delta\theta/\sigma_{e_k}$, in which $\dot{\psi}(r_k) = \partial\psi(r_k)/\partial r_k$, which is 1 for $|r_k| < h$, and 0 otherwise, from (B7). Substituting into (B9) gives

$$\mathbf{X}'^{\mathrm{T}}\mathbf{W}_e(\mathbf{W}_e\mathbf{I}'\,\mathbf{B}\,\Delta\theta + \psi(\mathbf{r})) = 0 \tag{B10}$$

dropping iteration subscripts, with $\mathbf{I}' = \mathrm{diag}(\dot{\psi}(\mathbf{r}))$ a modified identity matrix with its $kk$th element equal to $\dot{\psi}(r_k)$. Equation (B10) can then be rearranged to give $\Delta\theta$ at the current iteration

$$\Delta\theta = -(\mathbf{X}'^{\mathrm{T}}\mathbf{W}_e^2\mathbf{I}'\,\mathbf{X})^{-1}\mathbf{X}'^{\mathrm{T}}\mathbf{W}_e\psi(\mathbf{r}) \tag{B11}$$

Iteration proceeds until $\Delta\theta$ approaches 0.

The second iteration scheme is the iteration implemented in the python code. Such iterations are known to be stable, e.g. Maronna et al. (2019). However the logic of the first scheme is needed to obtain the covariance matrix of $\theta$, below. In the second scheme, a weight function, $w(r_k)$, is introduced so that the resulting equations have a simple least squares form. For (B9), this can be done by defining $w(r_k) = \psi(r_k)/r_k$, so that $\psi(r_k) = r_k w(r_k)$, introducing $\mathbf{r}$ explicitly into (B9). Then

$$\mathbf{X}'^{\mathrm{T}}\mathbf{W}(\mathbf{X}\theta - y) = 0 \tag{B12}$$

in which the $kk$th element of the diagonal matrix, $\mathbf{W}$, is $w(r_k)/\sigma_{e_k}^2$, combining the weighting from $w$ with the weighting from the data uncertainties, $\mathbf{W}_e$. By rearranging (B12), the repeated substitution solution to (B9) is given by

$$\theta = (\mathbf{X}'^{\mathrm{T}}\mathbf{W}\mathbf{X})^{-1}\mathbf{X}'^{\mathrm{T}}\mathbf{W}y \tag{B13}$$

Iteration is needed as $\mathbf{X}'$ and $\mathbf{W}$ depend slightly on $\theta$, given a sensible starting point for the iteration as provided by SIEGEL.

Accepting that an isochron has been calculated, the covariance matrix of $\theta$, $\mathbf{V}_\theta$, can be calculated by error propagation of the data to $\theta$ using the logic of the first iteration scheme. First it is convenient to transform the problem solved by (B11) to an identical one in which the data are represented by $\mathbf{e}$ rather than $\{\mathbf{x}, \mathbf{y}\}$. First note that (B9) involves $\mathbf{r}$, a scalar for each data point, the uncertainty on $\mathbf{e}$ in $\mathbf{W}_e$, and the $x$ position on the best-fit line where the ellipse around each datapoint touches the line, at $\mathbf{x}'$, the second column of $\mathbf{X}'$. Given the best-fit $\theta$, the $y$ corresponding to $\mathbf{x}'$, is $\mathbf{y}' = \mathbf{X}'\theta$. Defining $\mathbf{y}'' = \mathbf{y}' - \mathbf{e}$, data involving $\{\mathbf{x}', \mathbf{y}''\}$, rather than $\{\mathbf{x}, \mathbf{y}\}$, is an identical problem via (B9). In this identical problem, $\mathbf{x}'$ and $\mathbf{y}'$ are fixed (have no uncertainty). With this, the change in $\theta$ corresponding to (B9), becomes

$$\Delta\theta = -(\mathbf{X}'^{\mathrm{T}}\mathbf{W}_e^2\mathbf{I}'\,\mathbf{X}')^{-1}\mathbf{X}'^{\mathrm{T}}\mathbf{W}_e\psi(\mathbf{r}) \tag{B14}$$

in terms only of $\mathbf{X}'$ and not involving $\mathbf{X}$. This is akin to the transform in York et al. (2004) to convert the $\mathbf{V}_\theta$ of York (1969) to that of the $\mathbf{V}_\theta$ of Titterington and Halliday (1979).

Assuming that $\theta$ is approximately linear in each $e_k$ around the minimum in $\sum \rho(r_k)$, then

$$\mathbf{V}_\theta = \left(\frac{\partial \theta}{\partial \mathbf{e}}\right) \mathbf{V}_\mathbf{e} \left(\frac{\partial \theta}{\partial \mathbf{e}}\right)^{\mathrm{T}} \tag{B15}$$

At the solution of (B9), $\Delta\theta = 0$, so, by the chain rule using (B14)

$$\left(\frac{\partial \theta}{\partial \mathbf{e}}\right) = \left(\frac{\partial \Delta\theta}{\partial \theta}\right)^{-1} \left(\frac{\partial \Delta\theta}{\partial \mathbf{e}}\right) = (\mathbf{X}'^{\mathrm{T}} \mathbf{W}_\mathrm{e}^2 \mathbf{I}' \mathbf{X}')^{-1} \mathbf{X}'^{\mathrm{T}} \mathbf{W}_\mathrm{e}^2 \mathbf{I}' \tag{B16}$$

Substituting into (B15), cancelling and using $\mathbf{W}_\mathrm{e}^2 \mathbf{I}' \mathbf{W}_\mathrm{e}^{-2} \mathbf{I}' \mathbf{W}_\mathrm{e}^2 = \mathbf{W}_\mathrm{e}^2 \mathbf{I}'$

$$\mathbf{V}_\theta = (\mathbf{X}'^{\mathrm{T}} \mathbf{W}_\mathrm{e}^2 \mathbf{I}' \mathbf{X}')^{-1} \tag{B17}$$

The small steps in the age uncertainty (age$\pm$) curve in Fig A1 arise when diagonal elements in $\mathbf{I}'$ change from one to zero as
`mswd` increases.

In YORK (or if all $|r_k| < h$ in SPINE), then $\mathbf{I}' = \mathbf{I}$ and $\psi(\mathbf{r}) = \mathbf{r}$. So

$$\Delta\theta = -(\mathbf{X}'^{\mathrm{T}} \mathbf{W}_\mathrm{e}^2 \mathbf{X})^{-1} \mathbf{X}'^{\mathrm{T}} \mathbf{W}_\mathrm{e} \mathbf{r} \tag{B18}$$

and the covariance matrix becomes

$$\mathbf{V}_\theta = (\mathbf{X}'^{\mathrm{T}} \mathbf{W}_\mathrm{e}^2 \mathbf{X}')^{-1} \tag{B19}$$

If, in addition, all $\sigma_{x_k} = 0$ then $\mathbf{X}' = \mathbf{X}$, and iteration is not involved, $\mathbf{e}$ is replaced by $-\mathbf{y}$, and

$$\theta = (\mathbf{X}^{\mathrm{T}} \mathbf{W}_\mathrm{e}^2 \mathbf{X})^{-1} \mathbf{X}^{\mathrm{T}} \mathbf{W}_\mathrm{e}^2 \mathbf{e} \tag{B20}$$

with a covariance matrix of

$$\mathbf{V}_\theta = (\mathbf{X}^{\mathrm{T}} \mathbf{W}_\mathrm{e}^2 \mathbf{X})^{-1} \tag{B21}$$

These are the results for fitting data by simple weighted least squares.

**Appendix C: SPINE Python code**

The second iteration in Appendix B is coded in the Python function, `huber2`. The starting point for the iteration is provided by the function, `siegel` (Siegel, 1982). The calling function, `recipe`, is a placeholder for a more general function to be written by the user.

```
import sys
import datetime
```

```
      import numpy as np

      out = open("out.txt", "w") # opening output file
      screen = sys.stdout
standard = [screen, out]    # default where print goes

      defaulth = 1.4  # default h in huber

      def siegel(data): # siegel (1982)
n = data.shape[0];
          (x, sdx, y, sdy, cor) = np.transpose(data)
          x += 1e-8 * np.random.random(n) # naive breaking of x ties
          med = np.empty(n);
          for i in range(n):
col = np.empty(n);
              for j in range(n):
                  if i is not j: col[j] = (y[j] - y[i])/(x[j] - x[i])
              med[i] = np.median(np.delete(col, i))
          b = np.median(med)
return np.array((np.median(y - x * b), b))

      # -------------------------------------------------------

      def calcage(theta, covtheta = None, method = 1):
# tera-wasserburg: focus on lower intercept
          dc238 = 1.55125e-10; dc235 = 9.8485e-10
          tc = 0; t = t1 = 1500.; tdiff = 1e-4
          (a, b) = theta
          k = 0
while k < 12 and abs(t - tc) > tdiff:
              tc = t; t = 1/dc238 * np.log(1 + 1/t1)
              t1 = (1/137.8 * (np.exp(dc235 * t) - 1) / (np.exp(dc238 * t) - 1) - a) / b
              k += 1
          if covtheta is None:
sdage = 0
          else:
              den = b * np.exp(dc238 * t) * dc238 + \
                      (-np.exp(dc235 * t) * dc235 + np.exp(dc235 * t + dc238 * t) * dc235 + \
                       np.exp(dc238 * t) * dc238 - np.exp(dc235 * t + dc238 * t) * dc238)/137.8
jac = [(np.exp(dc238 * t) - 1)/den * (np.exp(dc238 * t) - 1), (np.exp(dc238 * t) - 1)/den]
              sdage = np.sqrt(np.dot(np.dot(jac, covtheta), np.transpose(jac)))/1e6
          return (t/1e6, sdage)

      # -------------------------------------------------------
```

```
      def huber2(data0, h = 1.4):
      #   iterativel-reweighted huber line-fitter
          n = data0.shape[0]
          itmax = 20; mindel = 1e-8; mincond = 1e-12; minsump = 0.01
(x, sdx, y, sdy, cor) = np.transpose(data0)
          avx = np.dot(x, np.ones(n))/n;   avy = np.dot(y, np.ones(n))/n;
          div = np.array([1/avy, avx/avy])
          data = np.copy(data0)
          (x, sdx, y, sdy, cor) = np.transpose(data)
x /= avx; sdx /= avx; y /= avy; sdy /= avy; cov = sdx*sdy*cor
          theta = siegel(data);
          k = 0; code = 0; deltheta = (1e10, 1e10)
          while k < itmax and ( np.sqrt(np.dot(deltheta,deltheta)) > mindel):
              k += 1; (a, b) = oldtheta = theta
e = a + b * x - y;   sde = np.sqrt(b**2*sdx**2 - 2*b*cov + sdy**2)
              r = e/sde
              wh = [1 if abs(rk) < h else np.sqrt(h/abs(rk)) for rk in r]/sde
              xp = x - r * (b*sdx**2 - cov)/sde              # x on the line
              ypp = (y - e - r*(b*cov - sdy**2)/sde)*wh      # W^(1/2)(y'-e): y off the line
c = np.transpose([wh, xp*wh])                  # W^(1/2) X'
              (u, s, v) = np.linalg.svd(c, full_matrices=False)
              if mincond * s[0] > s[1]: code = -1; break   # (nearly) singular matrix
              theta = np.dot(np.dot(np.dot(np.transpose(v), np.diag(1/s)), np.transpose(u)), ypp)
              deltheta = theta - oldtheta
if k == itmax: code = 1  # not converged
          pc = np.dot([wh**2, xp*wh**2], e);
          sump = np.sqrt(np.dot(pc, pc));
          if sump > minsump: code = 2  # not solved the nle
          dpsi = [1 if abs(rk) < h else 0 for rk in r]
covtheta = np.linalg.inv(np.dot(np.dot(np.transpose(c), np.diag(dpsi)), c)) / \
              np.array([[div[0]**2, div[0] * div[1]], [div[0] * div[1], div[1]**2]])
          return (code, theta/div, covtheta, sump, k, s[1])

      # -----------------------------------------------------
      def pr(s, e="\n", printto=standard):
      # prints a string
          for pr in printto: print(s, end=e, file=pr)

def recipe(title, data, where = [screen]):
      #   prototype calculation driver
          h = defaulth
          (x, sdx, y, sdy, cor) = np.transpose(data)
```

```
        n = data.shape[0]
today = datetime.datetime.now();
        pr("==========================================================\n"+ \
           "running spine.py on "+today.ctime())
        res = huber2(data)
        if res[0] != 0:
print("sample "+title+" not calculated")
            return "bad"
        (a, b) = theta = res[1]
        ucovtheta = res[2]
        (age, sdage) = calcage(theta, ucovtheta)
e = a + b * x - y
        sde = np.sqrt(b**2 * sdx**2 - 2 * b * sdx * sdy * cor + sdy**2)
        r = e/sde
        s = nmad(r); slim = 1.92 - 0.162 * np.log(10 + n);
        if s < slim:
pr(("sample "+title+": s = %0.2f"+ ": isochron " + "age = %0.3f +/- %0.3f Ma") % \
               (s, age, 1.96 * sdage), printto = where)
        else:
            pr(("sample "+title+": s = %0.2f"+ ": errorchron " + "age = %0.3f Ma") % \
               (s, age), printto = where)
return [0, age, sdage, theta]

    # ------------------------------------------------------

    # data rows:  x   sdx   y   sdy   cor
data = np.loadtxt("data0708.txt", delimiter=",")

    recipe("0708", data, where=standard); print()
```

Datafile for sample 0708, data0708.txt, see Fig. 6

```
    73.2064, 1.12543, 0.753, 0.0075, -0.068224
550 260.417, 4.06901, 0.435, 0.01, 0.18853
    169.205, 2.43357, 0.577, 0.01, 0.45644
    79.1766, 0.908995, 0.751, 0.01, 0.19585
    212.766, 2.94251, 0.473, 0.0085, 0.37494
    154.56, 1.79165, 0.615, 0.0105, 0.22567
555 217.391, 2.83554, 0.474, 0.0095, 0.27879
    209.644, 4.83455, 0.484, 0.011, 0.10969
    144.092, 2.49151, 0.647, 0.0105, 0.10222
    174.216, 1.97283, 0.584, 0.009, 0.21416
    224.215, 3.26771, 0.477, 0.012, 0.26464
560 236.407, 3.35329, 0.461, 0.0115, 0.14718
    161.551, 2.87086, 0.628, 0.011, -0.041164
```

```
265.252, 5.62869, 0.385, 0.0115, −0.043072
152.439, 2.32377, 0.608, 0.0095, 0.019543
151.745, 2.30266, 0.626, 0.0105, 0.0030752
```
```
101.112, 1.17572, 0.727, 0.0085, 0.038467
265.182, 3.41058, 0.427, 0.0095, 0.070384
286.78, 3.16634, 0.391, 0.0085, 0.24059
287.604, 3.5568, 0.365, 0.008, −0.041298
264.69, 2.83747, 0.419, 0.008, 0.15939
```
```
274.574, 2.67638, 0.378, 0.008, 0.17686
212.314, 4.28235, 0.501, 0.013, 0.34121
161.29, 1.69095, 0.591, 0.009, 0.14398
140.647, 1.58253, 0.633, 0.0105, 0.18753
183.15, 2.18036, 0.557, 0.0105, 0.49436
```
```
218.818, 2.39407, 0.53, 0.012, 0.21725
312.5, 5.3711, 0.334, 0.01, −0.080507
227.79, 2.20526, 0.49, 0.0105, 0.073412
212.766, 2.71616, 0.507, 0.011, 0.25544
139.276, 2.03676, 0.652, 0.01, 0.44912
```
```
179.533, 2.25625, 0.556, 0.0105, 0.24303
224.215, 2.51362, 0.48, 0.0085, 0.11165
219.78, 3.62275, 0.519, 0.0115, 0.31258
165.017, 2.8592, 0.59, 0.0115, 0.03575
145.773, 2.33746, 0.61, 0.0105, 0.098985
```
```
239.808, 4.02556, 0.442, 0.0095, −0.10478
187.266, 2.80548, 0.538, 0.0125, 0.11864
255.232, 2.63831, 0.443, 0.0085, 0.025538
141.443, 4.10124, 0.616, 0.011, −0.15512
115.34, 0.997753, 0.707, 0.01, 0.15087
```
```
117.509, 1.17371, 0.68, 0.008, 0.33891
73.6377, 0.623589, 0.757, 0.0065, 0.57883
160.256, 2.31139, 0.605, 0.012, 0.16382
149.701, 1.56872, 0.619, 0.009, 0.24589
245.278, 2.58694, 0.452, 0.008, 0.33748
```
```
251.256, 4.10343, 0.435, 0.014, 0.3589
130.208, 2.11928, 0.666, 0.011, 0.18376
276.243, 4.19706, 0.419, 0.01, 0.16668
298.508, 4.90087, 0.359, 0.0105, −0.090909
381.679, 6.40989, 0.241, 0.013, 0.0073965
```

Example output, running on the command line

```
running spine.py on Mon Sep 14 14:10:55 2020
sample 0708: s = 1.24: isochron age = 13.685 +/- 0.257 Ma
```

## Appendix D:  Simulation setup

This work was originally motivated by the dating of speleothems using the lower intercept with a U-Pb Concordia in Tera-Wasserburg style plots (Woodhead et al., 2012). This paper therefore discusses $\{x, y\}$ data with the expectation that $x =^{238}\mathrm{U}/^{206}\mathrm{Pb}$ and $y =^{207}\mathrm{Pb}/^{206}\mathrm{Pb}$, but the logic and the algorithm are in no way restricted to this system.

10,000 simulated datasets, each containing 5, 6, 8, 10 and 15 datapoints, respectively, were used to assess SPINE. Each dataset corresponds to an age of 4 Ma, with an underlying trend chosen to be $y = 0.811 - 0.000474737x$. For each dataset, the $x$-values were drawn from a uniform probability distribution with bounds, $\{400, 1100\}$ (so the $x$ are not equi-spaced). Datapoints are assigned uncertainty with $\sigma_{x_k} = 0$ and a fixed $\sigma_{y_k} = 0.00125$, the latter representing the analytical uncertainty, propagated from both the $x$ and $y$ measurement into $y$. In a real $\{^{238}\mathrm{U}/^{206}\mathrm{Pb}, \,^{207}\mathrm{Pb}/^{206}\mathrm{Pb}\}$ dataset, $\sigma_{x_k}$ and $\sigma_{y_k}$ would be finite and correlated. However, this makes no difference to the calculations once data are processed into $r_k$ form as in Fig. 2. For a given dataset, scatter is introduced into the data by drawing the $y$ values from an uncertainty distribution, centred on the underlying trend, that may be either Gaussian (N) or one of three contaminated Gaussian distributions—5%3N, 25%5N, or 10%10N—as in Powell et al. (2002). For $n = 10$ and gaussian-distributed uncertainties, the age uncertainty obtained is approximately $\sigma_t = 0.01$ Ma.

Results are presented in terms of kernel density estimates using an Epanechnikov kernel (Wand and Jones, 1995). Kernel density estimates (kde) are a way of presenting data that could otherwise be plotted as a histogram, normally normalised so that—like a probability distribution—the area under the kde curve is 1. The smoothness of the kde is controlled by a smoothing constant whose value was chosen to be just large enough for the kde to appear smooth, given that 10,000 datasets are used in each kde.

## Appendix E:  SPINE handling outliers

To see the consequence of outliers stemming from residuals from a contaminated Gaussian distribution, a dataset from simulations is shown in Fig. E1 and its corresponding quantile-quantile plot in Fig. E2. This simulation, with $n = 50$ and 25%3N and a true age of 4.00 Ma, follows the approach in Appendix D, but the uncertainty on the $y$ values is taken to be an order of magnitude larger (to be comparable with sample 0708). The shape of Fig. E2 is typical of contaminated Gaussian distributions in samples of this size, with outliers lying outside the band between the blue dashed lines. The SPINE age is 3.95 Ma, whilst the YORK age is higher, 4.11 Ma, $\Delta = +3.0\sigma$ of the SPINE age, due to the effect of the outliers. Mswd = 4.35, so ISOPLOT would give a model 2 age, 4.19 Ma, with $\Delta = 4.27$, being more affected by the outliers than YORK.

In general, with larger contaminated-Gaussian datasets, the difference between the SPINE and YORK ages are not dramatic, even if there are obvious outliers in the quantile-quantile plots. In 10,000 simulated datasets with $n = 50$ and 25%3N, a 95% confidence interval on $\Delta$ is $\pm 1.5$.

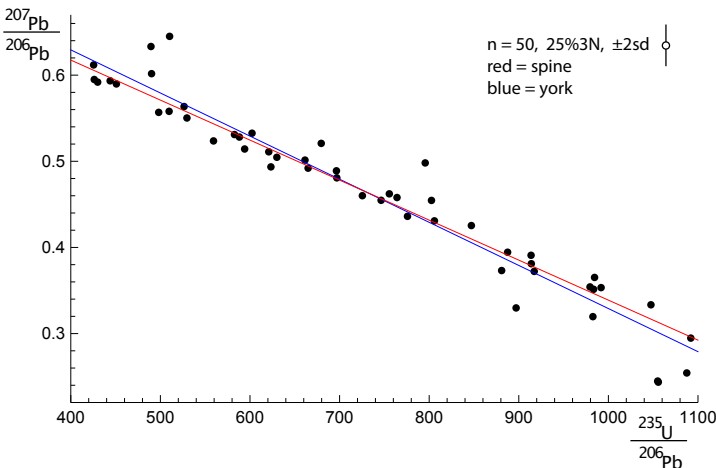

**Figure E1.** A 25%3N simulation with $n = 50$, with the SPINE fit in red, the YORK fit in blue. The $2\sigma$ uncertainty applied to the $y$ value of each datapoint is indicated in the top right of the figure. See text.

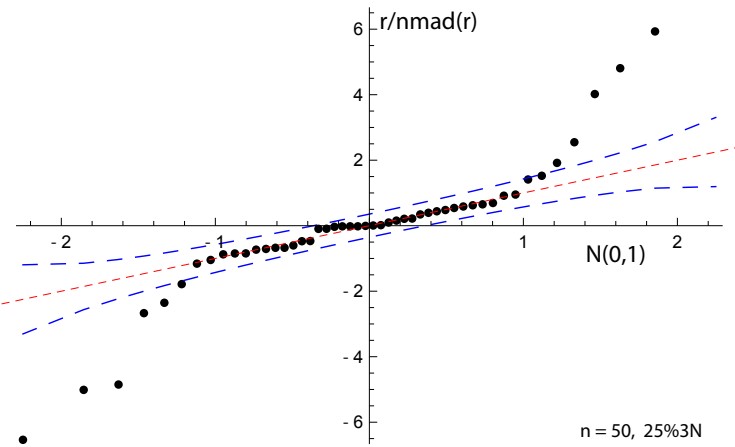

**Figure E2.** Quantile-quantile plot for example in Fig. E1. See text.

*Author contributions.* Roger Powell created the approach and coded the Python script; Eleanor Green helped validate the maths/statistics and write the paper; Tephy Marillo Sialer helped with the simulations; and Jon Woodhead oversaw the applicability of the approach.

*Competing interests.* The authors declare that they have no conflict of interest.

*Acknowledgements.* We would like to thank the anonymous reviewers for their work, particularly reviewer 4 (times two). Tim Pollard has materially helped in our understanding of the ISOPLOT code functionality. JW is funded by Australian Research Council grant FL160100028.

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
