# Peer review of "Robust Isochron Calculation"

_Geochronology, 2020_

## Referee Comment (RC1) · Anonymous Referee #1 · 17 Mar 2020

articleamsmathamsfontsamssymbgraphicx

**Comments on "Robust Isochron Calculation"**

The algorithm employed to compute the estimate will converge as long as the initial values are close enough to the solution. The algorithm is in this sense "correct", but it is unnecessarily complicated. I would prefer to use the algorithm of "Iterative weighted least squares", described in Section 4.5.2 of (Maronna et al, 2006), which is very simple and has guaranteed convergence.

As to the estimator itself, it has a flaw. Here the $x_k$'s are observed with a random error. This situation is called "regression with errors in variables". It is known that the Least Squares estimator, and in general any estimator based solely on the residuals –like the one proposed– has a bias that does not decrease with the sample size (it

is "inconsistent" in the statistical jargon). The adequate estimator for this situation is called "orthogonal regression", of which there are also robust versions. The size of the bias depends on the ratio between the standard deviation of the assumed $x$ errors (called $\sigma_{xk}$ in the paper) and that of the $x$'s. Unfortunately, for their simulations the authors choose $\sigma_{xk} = 0$ (line 247), which implies that there is no bias, and therefore this flaw is not visible in the results.

I cannot say however that the proposed estimator is "wrong": if the $\sigma_{xk}$ are "sufficiently small" compared to the dispersion of the $x$'s, the results may be accurate enough for practical purposes. How much is "sufficiently small" can be determined by more detailed simulations and/or theoretical calculations.

---

## Referee Comment (RC2) · Anonymous Referee #2 · 19 Mar 2020

General comments

This paper discusses the use of robust regression estimators in the context of isochron calculations. It seems that this is a timely and useful contribution, and I believe this is a good first paper on the topic. I am particularly glad to see the authors include their code in the submission.

Specific comments

Since I am not familiar with the model, it took me several readings to realize that $r_k$ did not denote regression residuals ($y_k - a - b * x_k$), but rather standardized residuals where the sd is itself a function of the parameter b (c.f. equation A3). This necessarily complicates the calculation, and algorithms designed for "standard" regression/scale models do not apply here. I would recommend that be discussed explicitly earlier in the manuscript, to highlight that this is not just a simple linear regression problem.

[Figure]

My main two suggestions are: (a) Figure 6 seems to suggest that high-leverage points may be present in the data. In that case, it is known that Huber-type loss functions may not protect against this type of outliers. A re-descending rho function is needed in such situations, and I would suggest that the authors mention this, and maybe run a small numerical experiment to assess its potential advantage. (b) The covariance matrix $V_k$ (in equation (A1)) can also be affected by atypical observations. It may be a good idea to replace it with a robust alternative.

Technical corrections

The authors may want to refer to the second edition of this book: Maronna, Martin, Yohai and Salibian-Barrera, Robust Statistics: Theory and Methods (with R), 2nd Edition, Wiley, 2019.

---

## Author Comment (AC1) · 14 Apr 2020

As noted by the reviewer, the regression attempted is not simple linear regression, as is explicit in the development of YORK. This is a given in the standard isochron calculations that the current manuscript intends to augment or supercede.

Suggestion (a) of the reviewer is certainly relevant and is now considered in the Discussion of the manuscript. High leverage data particularly in small datasets is an unresolved problem. As covered in Maronna et al. (2019), ch. 5, there are several difficulties in general, the first being what to use as $\rho$ in the estimator, as in Appendix 1 of the manuscript. Whereas a HAMPEL (a re-descending $\rho$) is advocated in Maronna, this is inadvisable in small datasets, not wishing to "lose" information. A second, bigger problem is what to use as the initial estimate of the linear trend—prior to using this $\rho$—given that the $L1$ estimate used in our algorithm is not robust in the presence of such

data. Although Maronna et al. (2019) make detailed suggestions, it is not clear how they could be married with using the analytical uncertainties in the algorithm. A way forward is simply to follow Huber and Rochetti (2009), e.g. their p. 150 and p.161, who note that trying to automate safeguarding against high leverage data is overrated, and that it is better to rely on diagnostics (e.g. `hat` to identify high leverage) and human judgement in data assessment.

Relating to suggestion (b), (A1) is simply the uncertainties for datapoint $k$ in covariance matrix form, derived from the analytical measurements. It is not clear what is being referred to by the reviewer here? The newer Maronna is now cited, an edition we were unaware of (thank you).

---

## Author Comment (AC2) · 14 Apr 2020

The reviewer suggests that an "iterative weighted least squares" algorithm (e.g. Maronna et al, 2006, Section 4.5.2) should be preferred. In the process of finding an algorithm for our study, we did initially devise an iterative weighted least squares algorithm that uses the analytical uncertainties as the scale of data scatter, rather than the usual robust regression scale given by the scatter of the data about the linear trend. But in fact the algorithm converged only very slowly (100s of iterations), making it impractical. The algorithm eventually adopted may not converge for "poor" data from a less good starting estimate, but in our experience it works well (with less than 5 iterations for the family of contaminated-Gaussian datasets investigated—and showed no failures to converge in application to hundreds of thousands of simulated datasets). Additions to the algorithm may be needed to allow handling of poor data but such handling was not part of the object of our study. Contaminated-Gaussian datasets are relatively

well-behaved (they are relatively "good" datasets), even though `mswd` may be large. Indeed, most datasets that geochronologists would feed into an isochron calculation also tend to be classified as relatively good.

We are confused by the idea that YORK and by extension the approach taken in our manuscript is "inconsistent". The algorithms work on the residuals (A4). Each is a scalar for a datapoint, given the data and analytical uncertainties. Although the data may involve uncertainties on $x$ and $y$, a particular residual could be considered as just an uncertainty on $y$, with the uncertainty on $x$ being zero. There is nothing in the scalar which flags there is, or is not, an uncertainty on $x$. In this sense, is YORK really "regression with errors in variables", given that the uncertainty on each data point is specified by its analytically-defined covariance matrix (A1)? Regardless of this, it is certainly true that the design width in isotopic datasets is generally much bigger than the individual $x$ uncertainties, so according to the reviewer's explanation, the problem alluded to would be minimised.

---

## Referee Comment (RC3) · Anonymous Referee #3 · 16 Apr 2020

This manuscript examines regression approaches used in isochron age calculations. As a frequent user of regression calculations that are part of determining isochron ages, I welcome a renewed discussion on this topic. However, I am not a statistician and so my review will be of little use regarding the nuts and bolts of this paper and not in a position to judge the scientific significance and quality. Rather, I will comment on how effectively (or not) the authors are communicating the essence of their results. One measure of a paper is how well it communicates its message to its intended audience. I recognize that mainly specialists—those interested in the statistics of isochron and age calculations—will be the (small) handful of readers of this paper, but it would benefit the practitioners in this field as well if it was made more accessible. There are a few ways to do this. First, throughout this article, the writing is very much filled with jargon, making it difficult for the general reader to access the meat of what the authors have to say. This is especially important in the introduction where the authors should

communicate more clearly why their work is important. Why would we want to use HUBER approach, rather than YORK? Are the calculated dates themselves different? The uncertainties? Whether or not the regressions meet the definition of an acceptable date/isochron? These questions are not at all well raised, much less effectively answered, in this manuscript. If the authors want their work to have impact, they need to do a much better job of communicating and making their work more accessible to a broader audience.

Other comments:

The nature of a manuscript like this is to include many formulas. I get that, but at the same time, all of the terms in the formulas need to be defined. This is also true for the tables—they need to be explained more fully to communicate their information effectively with the reader.

The writing in general is very stiff and not easily digested. Part of this is the use of too much jargon, but it could use "softening" and significant wordsmithing throughout. Perhaps some of the coauthors could help with this.

Small, but important point: From the beginning the ms refers to the mswd parameter to define goodness of fit and whether or not a regression passes or fails the mswd test. It would be useful to have a general discussion at the beginning of what defines "passing" or "failing".

In summary, I am not able to comment on how robust the statistical treatment discussed here is, but I do feel that the ms should be made more "accessible" to a broader audience.

---

## Referee Comment (RC4) · Anonymous Referee #4 · 23 Apr 2020

[11pt]extarticle[]graphicx[]color

framed

alltt [T1]fontenc [latin9]inputenc geometry verbose,tmargin=1in,bmargin=1in,lmargin=1in,rmargin=1in float amsmath amsthm setspace [unicode=true,pdfusetitle, bookmarks=true,bookmarksnumbered=true,bookmarksopen=false, breaklinks=false,pdfborder=0 0 1,backref=false,colorlinks=false] hyperref

[bottom]footmisc

[style=authoryear,natbib=true,firstinits=true,backend=biber]biblatex

sortnamelast-first

inflFncs.bib

txfonts upgreek

upquote

**Review of "Robust Isochron Calculation" by Powell et al. (gchron2020-4)**

April 23, 2020

**1  Summary**

Prior to writing this review I have read the reviews by the three Anonymous Reviewers AN1, AN2, AN3, abd have also looked over a number the paper's references. The application of robust regression straight line fits with errors in variables to isochron calculation is an interesting one, and it seems potentially important in analyzing such data. It is striking how sketchy the body of this paper is compared with the very well written and more detailed treatment of robust regression applied to isochron calculation by Powell et al. (2002). One concludes that the main contribution of the current paper is the computational algorithm in Appendix A, and that the paper is aimed at a very small audience of specialists who are quite familiar with the paper's main reference. I could therefore just comment on the computational algorithm. But in case the authors and the journal want to a paper that is accesible by a wider audience, in which case reviewer AN3's comments are relevant, I offer the comments below with that possibility in mind.

**2  Improvements Needed to Serve a Wider Audience**

**2.1  Empirical Motivation**

The use of robust statistical methods to deal with non-normal distributions associated with outliers is important. But the first step in justifying a paper that is applications

oriented, as is this one, is to provide empirical evidence demonstrating that the chronic data indeed exhibits such characteristics. This should minimally be done in Section 3 for the data of Figure 6, by making a normal qqplot of the residuals from the linear fit. But, it would be better to do it at the beginning of the paper, in one way or another, for example: (a) display a qqplot of the residuals from the ultimate fit (point to Section 3 with regard to the fit), or (b) using residuals from an exploratory robust fit with a repeated medians estimate, as discussed in Siegel, (1982). "Robust Regression Using Repeated Medians". *Biometrika*, 69, 1, 242-244.

**2.2   The Errors in Variables Problem**

As was pointed out by AN1, the authors are trying to compute a robust straight line fit taking into account that their problem is an errors-in-variables (EIV) problem. While the authors do not state this clearly, it is immediately obvious from the paper's two York references and McClean reference that the focus is on an EIV problem. The authors need to provide up front the basic equations that describe the EIV problem and state the well-known results that classical least square (LS) estimate is biased (stating the bias formula), unless there is no error in the independent variable, and furthermore this bias does not go away as the sample size goes to infinity. (a "consistent" estimator is one that converges to the true parameter value, in a probabilistic sense as the sample size goes to infinity).

The main place that one sees clearly that the paper is trying solve a classic EIV problem is in equations (A1) - (A4) of the Appendix. But then in order to solve the problem, the values of the variances $\sigma_{x_k}^2$ and $\sigma_{y_k}^2$ and the correlations $\rho_{x_k y_k}$ need to be specified. One can infer from the comments in Section 2.5 and the ellipses in Figure 6, that those values were used for each $x_x, y_k$ pair, but there is no substantive discussion of this. For the sake of reproducible research, the authors should make available the data set and values used for those variances and correlations.

However, based on Figure 6, one might assume that the $\sigma^2_{x_k}$ are relatively small enough for that data to be ignored and just use classical LS (no EIV). In fact the authors should compare the their result with that of classical LS. Of course for other chronic data sets the errors in the independent variable may be much larger?

**2.3 The Explanation in Section 2.2**

Like reviewer AN2, it took a repeated reading to figure out that the $r_k$ are standardized versions of the regression residuals $e_k$. The explanation in Section 2.2. is quite confusing in several ways, the first of which is the sentence "The residual for data point, k, used, denoted $r_k$, is the scaling factor . . . ". It seems clear from reading Appendix A, that what is meant is that the $r_k$ are the scaled residuals given be (A4), where the numerator $e_k$ there is the raw residuals (except the sign should be changed in the numerator), and the denominator is the weight given to the residual to take care of the errors-in-variables aspect. The writers should just state this clearly in the body of the text, by properly introducing the EIV problem and how the denominator scaling is actually computed (one can infer this in Appendix A, but readers will appreciate a clear up-front explanation).

The sentence in the last line of page 3 states: "Although it is not obvious from the form of $\rho(r_k)$, HUBER is equivalent to bringing data points in to $\pm h$ when $\mid r_k \mid > h$ .", is indeed not obvious. But the authors should make it obvious by plotting the shape of the $\psi(r_k) = \rho'(r_k)$, and emphasizing that $r_k = e_k/\sigma_{e_k}$ , where that behavior will be totally obvious.

**2.4 Concerning the mswd Criterion**

The mswd is finally defined in equation (2), and I think the paper would be better served to introduce it early on in connection with a very brief but clear review of the nonlinear
LS EIV minimization problem statement. In any event the equation (2) should have a right-hand side where $r_k$ is replaced by $e_k/\sigma_{e_k}$ to continue to emphasize that the $r_k$ are the standardized residuals. This is quite in keeping with equation (1) of Wendt & Carl (1991).

In equation (3) I think that the authors mean that **nmad(r)** is the standardized median of the **mad(r)**, the latter of which is more clearly written as $mad\{r_k\} = med\{|r_k - med\{r_k\}|\}$, and provide the normalization constant 1.4826. This is a very good thing to do, i.e., use a robust scale of the standardized residuals $r_k$.

If my assumption above is correct, the authors should reconsider the sentence "Given that s is based on a median, its magnitude depends on that half of the data that have the smallest absolute values of r", maybe modify it, maybe delete it.

2.5  Statistical Efficiency and Choice of Huber Psi Function Parameter

The terms "efficiency" first appears in the second line of Section 2.2 and the term "efficient" appears in the next to last paragraph of Section 2.2, and both appearances are followed by the phrase "(see below)". Based on that, I expected later on to see a definition of efficiency that is commonly used in statistics , and in robust statistics in particular. But the discussion of efficiency that finally occurs in Section 2.4 does not provide such a definition, and is not of little use. The efficiency (EFF) of a robust estimator $\hat{\theta}$ at a data distribution $F$ is defined as that ratio of the variance of a "best" estimator at data distribution $F$ to the variance of the robust estimator at data distribution $F$

$$\text{EFF}(\hat{\theta}; F) = \text{var}(\hat{\theta}_{best}; F)/\text{var}(\hat{\theta}_{robust}; F)$$

usually expressed as a percent. In the context of the manuscript, the data distribution $F$ is taken to be a normal distribution and the best estimator for a normal distribution is the

LS estimator. It is common practice to choose the tuning constant, i.e., $h$ in the paper so that the estimator has 95% normal distribution efficiency. And there is a trade-off between normal distribution efficiency and robustness toward non-normal distributions that result in fat-tailed distributions and outliers. One can achieve more of the latter robustness by decreasing the normal distribution efficiency, and vice versa. I provide this level of detail because it is important with regard to the comments I make **below**.A an asymptotic version of efficiency of a robust estimator, in the context of the trade-off between normal distribution efficiency and robustness, can be found in Section 3.4 of Maronna et al. (2019).

2.6   Focus on Normal Mixture Distributions

A focus on specific normal mixture distributions is a seemingly natural thing to do. But it is not of much help unless the researcher is pretty certain the the data (the error term in the straight line regression model in the paper) really conforms to such normal mixture distributions, based on a sufficiently large data set size. This could be easily checked by examining normal qqplots of the residuals from the fit. Two component normal mixture models that generate outliers results in a normal qqplot with two approximately linear pieces, and often one finds three linear pieces that require a three component normal mixture models. The authors could easily check this for the residuals in their real data example.

If such analysis does not indicate that a normal mixture distribution is reasonable, then there is no point in focusing on a a normal mixture distribution.Furthermore, the point of robust estimation is that for data whose distribution has a central region that is normal, the variation in the tails and extent of outliers does not have much influence on the variability of the robust estimator.

[Figure]

**2.7 Choice of Robust Loss Function and Associated Psi Function**

As Powell et al. (2002) clearly indicates, the lead author of the current paper (and possibly co-authors) is quite familiar with some of the basic robustness material in Hampel et al. (2006), including the possibility of using not only the Huber psi function, but also a redescending psi function. The focus on the Huber psi function in the current manuscript is quite reasonable both intuitively and theoretically. With regard to the latter, the Huber psi function has a well-known optimal property of (asymptotically in sample size) minimizing the maximum variance over a mixture model where the central model is normal and contamination component can be any symmetric distribution (which therefore does not induce estimator bias). Furthermore , the optimization problem is convex in this case. The tuning parameter $h$ in the present manuscript is conventionally chosen to achieve a specified normal distribution efficiency, typically 95%.

On the other hand, under the more realistic model that the contamination component is completely unrestricted (and therefore can lead to estimator bias), there is an optimal bias robust regression estimator, whose loss function is bounded (unlike the unbounded Huber loss function) and whose psi function smoothly redescends. This estimator is described in Section 5.8.1 of Maronna et al. (2019). This estimator also has a tuning parameter that is set at a user's desired normal distribution efficiency, again typically at 95%. Although the optimization problem is no longer convex, there exists a very good algorithm for computing the optimal bias robust estimator for a specified normal distribution efficient. This method is available in the R package RobStatTM available at CRAN uses that algorithm.

For the kind of data in Figure 6 of the manuscript, the Huber choice should do well since the independent variable leverage is not very large, but the optimal bias robust regression may do better and would be worth trying. For other kinds of data sets with large leverage outliers or very fat-tailed residuals, the optimal bias robust estimator may

provide a better solution. However, adapting this estimator to the EIV problem solved by the algorithm in Appendix A would be challenging.

Minor Comment

It seems quite inappropriate to use HUBER to refer to the M-estimator (Maximum-likelihood type estimator) using his favorite rho and psi function. It would be better to use M-estimator is the basic concept, and say that the paper focused on the Huber rho and psi. Similarly, use of YORK to refer to the algorithm he published seems awkward, particularly since he was not the only person to introduce that algorithm.

2.8   The Computational Algorithm

It seems that computation algorithm in Appendix A is the main contribution of the paper. While the Huber M-estimator optimization problem is convex and the numerical algorithm is known to converge, this EIV formulation is no longer convex, and though it may work well, the iterative algorithm might converge to a bad local minimum, or possibly fail to converge. It would be good to have a convergence proof, but that may be exceedingly difficult or impossible. In any event a good starting point helps a lot, and for the general multi-factor regression the L1 start is a good one.

However, for the simple straight line regression problem, the Siegel (1982) repeated median estimator, available in the R package `mblm` is likely to be even better as an initial condition for the Appendix A algorithm (the repeated median has low efficiency but has a breakdown point of 1/2, as compared with the L1 breakdown point of zero). One thought about the possible use of the repeated median estimator is the following. Use the repeated median estimator to define the slope needed in (A3) to define the weighted residuals in (A2), and hold it fixed while solving the robust regression problem. In this case you can use any of the robust regression computational methods available
in the R package RobStatTM, including robust regression based on the Huber psi or the optimal bias robust psi mentioned int he previous subsection. And one can then do a few outer loop iterations to update the fixed values for (A3), and maybe even one or two iterations would suffice.

---

## Author Comment (AC3) · 3 May 2020

[ The response-to-review lodged earlier was before we knew that 2 more reviews would be forthcoming. The first paragraph of this response is largely unchanged, but the second paragraph is new, replacing the original second paragraph. This change is a consequence of the 4th review and the reading done in response to that. ]

The reviewer suggests that an "iterative weighted least squares" algorithm (e.g. Maronna et al, 2006, Section 4.5.2) should be preferred. In the process of finding an algorithm for our study, we did initially devise an iterative weighted least squares algorithm that uses the analytical uncertainties as the scale of data scatter, rather than the usual robust regression scale given by the scatter of the data about the linear trend. But in fact the algorithm converged only very slowly (100s of iterations), making it impractical. The algorithm eventually adopted may not converge for "poor" data and from

a less good starting estimate, but in our experience it works well (with less than 5 it-erations for the family of contaminated-Gaussian datasets investigated—and showed no failures to converge in application to hundreds of thousands of simulated datasets). Additions to the algorithm may be needed to allow handling of poor data but that was not part of the object of our study. Contaminated-Gaussian datasets are relatively well-behaved (they are relatively "good" datasets), even though $\mathtt{mswd}$ may be large. Indeed, most datasets that geochronologists would feed into an isochron calculation also tend to be classified as relatively good. See also the response to review 4.

The reviewer, not realising that the algorithm in the manuscript (and indeed YORK) properly accounts for errors in $x$ as well as $y$, clearly thought we had fallen into the error-in-variables trap. If we had fallen into that trap, the slope estimates from the algorithm might indeed have been biased downwards, and the approach would have been inconsistent. Section 2 in the manuscript now provides the background necessary for such a reader to more clearly see that the algorithm is sound in that regard. See also the response to review 4.

---

## Author Comment (AC4) · 3 May 2020

[ The response-to-review lodged earlier was before we knew that 2 more reviews would be forthcoming. The main change to this response is the addition of the 3rd paragraph ]

As noted by the reviewer, the regression attempted is not simple linear regression, as is explicit in the development of YORK. This is a given in the standard isochron calculations that the current manuscript intends to augment or supercede. Section 2 of the manuscript has been expanded to bring out this idea; see also the response to review 4.

Suggestion (a) of the reviewer is certainly relevant and is now considered in the Discussion of the manuscript. High leverage data particularly in small datasets is an unresolved problem. As covered in Maronna et al. (2019), ch. 5, there are several

difficulties in general, the first being what to use as $\rho$ in the estimator, as in Appendix 1 of the manuscript. Whereas a HAMPEL (a re-descending $\rho$) is advocated in Maronna, this is inadvisable in small datasets, not wishing to "lose" information. A second problem is what to use as the initial estimate of the linear trend—prior to using this $\rho$—given that the $L1$ estimate used in our algorithm is not robust in the presence of such data. The suggestion in review 4 of using Siegel's repeated median, not $L1$, might resolve this problem. Although Maronna et al. (2019) make detailed suggestions in their Section 5.8.1 for handling high leverage data, it is not clear how they could be married with using the analytical uncertainties in the algorithm, as also noted in review 4. A way forward is simply to follow Huber and Rochetti (2009), e.g. their p. 150 and p.161, who note that trying to automate safeguarding against high leverage data is overrated, and that it is better to rely on diagnostics (e.g. `hat` to identify high leverage) and human judgement in data assessment.

The datapoint at highest $x$ in Fig. 6 is pointed to by the reviewer as a high leverage point. If this datapoint is omitted, then HUBER gives $13.75\,\mathrm{Ma}$ rather than $13.69\,\mathrm{Ma}$. The difference normalised to the standard deviation on the age is only 0.47. The datapoint might be omitted only if there is a good geological reason to do it. This example is also included in the Discussion now.

Relating to suggestion (b), (A1) is simply the uncertainties for datapoint $k$ in covariance matrix form, derived from the analytical measurements. It is not clear what is being referred to by the reviewer here? The newer Maronna is now cited, an edition we were unaware of (thank you).

---

## Author Comment (AC5) · 3 May 2020

The manuscript was not written for a reader like reviewer 3! There is a collision between what this reviewer would like and what, e.g. reviewer 4 would like, but clearly in the original manuscript we did not meet the needs of either cohorts of readers. We accept that the manuscript should be made more approachable and we have now tried to make it so, following the suggestions of this reviewer, as well as reviewer 4.

It is disappointing that the main aim of the work was invisible to reviewer 3. This aim is spelt out in several places in the original manuscript, including the Introduction and the Discussion—that HUBER allows many more datasets to be called isochrons than classical methods do. This message is now expanded in the Introduction, as asked for by the reviewer, and as well relevant background material is added there. Also we do see now that our case can be, and now is, made more strongly. In particular, there is no

discontinuity in calculation method with increasing excess scatter as in ISOPLOT, and the ages gained from the additional datasets that are isochrons under HUBER are more reliable than they would be if calculated with YORK. For example, for new simulations of 10,000 datasets with $n = 10$ and scatter from 10%10N, using the approach in Appendix B, such additional datasets show that the 95% confidence limit on the ages is 3.97 to 4.03 Ma with HUBER, but 3.91 to 4.09 Ma under YORK, a significant increase in reliability with HUBER.

The idea behind `mswd` is now explained better, and what is meant by "pass"/"fail". As far as we are aware all symbols are defined when they are first used, as is the convention, and what the reviewer calls jargon are mainly technical terms that are needed for clarity? Regardless, we hope that all the changes to the manuscript will allow readers like reviewer 3 to make progress understanding what we are suggesting.

---

## Author Comment (AC6) · 3 May 2020

As noted in the review's summary, the algorithm and the code is an important part of the manuscript, so the reviewer's comments on that will be discussed first. But we would like the manuscript to be more accessible too.The reviewer's comments on improving accessibility are appreciated. As noted in the response to review 3, there is a difficulty making the manuscript more accessible to both geologists and statisticians, but we have tried to do this.

Regarding the computational algorithm, it is recognised that the algorithm might not converge, and the code flags if that happens. The logic to find a good starting point—now discussed in the Appendix—is that several different estimates are calculated initially, including $L1$. The estimate with the smallest $\sum \rho(r_k)$ (the sum being minimised in the algorithm) is used as the starting point. The HUBER function allows the user to

call it with their choice of possible additional starting points. Following the reviewer's suggestion, the Siegel repeated median estimator has been added to the calculations that the HUBER function does to find a good starting point.

The reviewer's suggestion for an improved algorithm is certainly interesting and will be investigated in due course. But we can refer here to our response to review 1: in our experience the algorithm works well, with less than 5 iterations for each simulated dataset in the family of contaminated-Gaussian datasets investigated—and showed no failures to converge in application to hundreds of thousands of such datasets. It is worth repeating, too, that the greatest majority of actual isochron datasets are not chronic, for better or worse having been cleaned of more gross outliers by users.

Regarding improving accessibility, the parts of Section 2 in the review will be considered in turn

2.1: As already noted in the Introduction to the manuscript, testing for the distribution of the scatter in the data is not possible, but with insufficient detail why. The fact is that the greatest majority of geochronological datasets are of the order of 10 datapoints.

The dataset in Fig. 6 is unusual in being so large. The `qqplot` for this appears to be in the range of strictly Gaussian datasets of 50 datapoints, from running several simulations. With the small excess scatter as reflected in the relatively small `mswd`, *more* datapoints in a dataset would be needed to ascertain if it is a Gaussian mixture of the sort simulated in the manuscript.

2.2: With current isochron calculations stemming from the work of York, and the first author's lack of knowledge of that part of the stats literature, he was unaware of the Error-in-Variables/Measurement-Error work. These terms have not been used in motivating isochron calculations in the geochronological literature, though the main idea underpins the Introduction in York (1966). Via Fuller (1987), this

aspect is now covered in the manuscript, allowing readers, including statisticians, to see that link. Unlike many measurement error problems, uncertainties in $x$ can be properly accounted for in isochron calculations as they are known from the analytical work generating the isotopic data.

The data for Fig. 6 are now included in Appendix C, following the Python code. Fitting the data just with ordinary least squares and data uncertainties from the data scatter, as in Model 2 calculations, has a slightly smaller slope than YORK possibly indicating a small downward bias, but just with a difference of -1.2 $\sigma$ on the slope.

2.3–2.5: Section 2.2–2.3 of the manuscript has been rewritten to cover what the reviewer is suggesting. Clearly it was a mistake to sweep as many of the equations as possible into the Appendix! Efficiency is now explained much better in the text.

2.6: As noted above, the greatest majority of geochronological datasets are of the order of 10 datapoints, so that checking for the form of excess scatter is not possible. Investigating the family of Gaussian mixture distributions seemed like a natural thing to do, as the reviewer agrees?

2.7: The possibility of using a redescending $\rho$ function, and also the other methods advocated in Maronna (2019), Section 5.8.1, is discussed already in the response to review 2. At least reviewer 4 ackowledges that adopting such methods for the error-in-variables calculations in the manuscript would be challenging

2.8 minor comment: Using YORK and HUBER as the names for the two main approaches discussed in the manuscript does have the merit of simplicity. The words around choosing these terms can be improved easily. We are at the mercy of the subeditor here.

---

## Author Response (AR1)

**response to associate editor**

1. Clearly there is a difficult balance to strike between accessibility and providing full details of our new approach. We have moved much of the mathematical formulation into Appendices, but retain key information in the main text. We have also tried to provide insight into the (often cryptic) protocol and calculations implemented in the widely-used ISOPLOT software of Ken Ludwig.

2. In order to address the associate editor's suggestion, a comparison of application of different algorithms, including those used in ISOPLOT, is included in the revision in a new Appendix, using the data used for the old Fig. 6 (now Fig. 7). These data are now included in Appendix C as requested by reviewer 4, making it straightforward for the reader to see and assess the results.

The robust algorithms in Ken Ludwig's ISOPLOT are given in the isoplot manual 3.75, (2008 revision) p. 25, with details in the papers cited. Strictly these are resistant algorithms, having high breakdown point, but low efficiency (e.g. Huber & Rochetti, 2009, Sect. 1.2.3). In our algorithm, and the code in our manuscript, such methods are used to provide a *starting point* for the iteration in SPINE, as now spelt out at the start of Appendix C. Including Siegel (1982) was the idea of reviewer 4, and that has been implemented in the code and included in the manuscript.

Ken Ludwig was a reviewer of the Powell et al. (2002) paper, and was enthusiastic about the work. Subsequent communications between Ken and the 1st author were aimed at including the robust isochron calculation approach of the 2002 paper in ISOPLOT. Unfortunately this didn't happen, due to personal matters that Ken experienced immediately after that time. In the ISOPLOT documentation regarding robust regression, his first suggested method is superseded by the second (Siegel, 1982). Looking at Ken's vba ISOPLOT code, the bootstrap used with the Siegel (1982) data-fitting is a simple "cases", or structural-based resampling, as though the $x$-$y$ data are drawn from some underlying (unknown) bivariate probability distribution (not the more appropriate, model-based approach used in the 2002 paper). However, any bootstrap is difficult and undeveloped for robust regression, e.g. Davison & Hinkley (1997), Sect. 6.5, and unreliable in general for median-based calculations. Significant effort went into developing the bootstrap for our current work - without success. In the 2002 paper the analytical uncertainties were discarded before the regression and the bootstrap were calculated, as already stated in the Discussion of the manuscript. Integrating the analytical uncertainties into the bootstrap contributed significantly to our difficulty.

3. We think that there may be a misunderstanding here regarding the sentence quoted from our response to review 3, "...such additional datasets show that the 95% confidence limit on the ages is 3.97 to 4.03 Ma with SPINE, but 3.91 to 4.09 Ma with YORK, a significant increase in reliability with SPINE". This is a 95% confidence interval on *all* the individual ages of such datasets, 4445 of them, that fail mswd but have acceptable spine width, using the cutoffs in Table 1 in the manuscript. Reliability is possibly not the best word to use, but this result flags how much better SPINE does than model 2 (or YORK) in this simulation.

Philosphically isochron-errorchron and model 3 are orthogonal to each other. In the former, a general purpose calculation method is sought which includes a way to distinguish datasets that are more likely to have age significance (isochrons) from those which are less likely (errorchrons). In the latter, data scatter is parameterised with analytical uncertainties and additional contributions depending on the geological processes envisaged to be involved. In the latter, maximum likelihood can then be used to devise an algorithm to find the model parameters, if the imagined data uncertainty structure is completely specified. We, in our manuscript, aim for the former, whereas the reviewer has focused on the latter in recent publications.

Model 3 calculations can be very interesting when there are geological reasons to make them plausible. However, commonly, identifying the cause of excess scatter is not easy to do or may indeed be impossible. Instead, our approach aims to calculate age information, discounting the potentially deleterious effects of excess scatter. Using a robust statistics logic, isochron calculations can be undertaken for a wider range of datasets. Importantly our approach is consistent with YORK for "good" datsets (e.g. with mswd less than some cutoff, as in model 1, accepting that the data (analytical) uncertainties account for data scatter). But, in addition, the approach extends seamlessly to datasets where mswd is

greater than the cutoff (e.g. model 1 with expanded errors, as well as model 2). This extension is considered to also give isochrons as long as the central spine of the data, $s$, is less than some cutoff. The cutoffs, depending on the number of datapoints, are given in Table 1 of the submitted manuscript, from a 95% confidence interval on $s$ for simulated datasets with Gaussian-distributed data uncertainties. The use of these cutoffs is directly analogous to the use of cutoffs on `mswd`, also given in Table 1. The use of such $s$ cutoffs avoids the rubbish-in rubbish-out problem, in the same way that `mswd` cutoffs do in YORK fitting, for strictly Gaussian data uncertainties.

We would contest the idea that maximum likelihood and the tests that can be formulated in that framework are always that useful, particularly for the small size of datasets that are typical of geochronology. As noted already in the response to review 4, even the 51-point dataset of Fig.6 is insufficient to distinguish a strictly Gaussian data uncertainty structure from a contaminated Gaussian one. In YORK calculations the adoption of a strictly Gaussian uncertainty structure is an assumption - it cannot be tested for. It is well known that even trivial departures from this assumption have a deleterious effect on the results from using classical statistical methods (see, for example, the introductory chapters of Huber, 1982, and Hampel et al., 1985). This may well apply in model 3 calculations too. Whether the assumptions in the parameterisation cause a problem in algorithms used for model 3 calculations could easily be tested with simulations, for example using very slightly contaminated Gaussian data, e.g. 5%3N.

Certainly, in the current manuscript we are not following the "cult of `mswd`". Far from it. We are advocating that this statistic is not useful and we do not use it. `Mswd` appears in the manuscript simply because it is needed to show that SPINE subsumes YORK for "good" datasets. Indeed, because of that, our algorithm is based on a minimis- ation that reduces to minimising `mswd` for "good" datasets. Of course, `mswd` is pivotal in model 3 calculations given that it is used to signal the existence of excess scatter. But, if, as suggested in Vermeesch (2018), excess scatter starts at $\texttt{mswd} = 1$ with increasing`mswd`, rather than at some cutoff from a confidence interval on `mswd`, then a model 3 calculation might well be trying to model something which is not there statistically, just noise? See Table 1 in the manuscript in the context of Appendix A, discussing ISOPLOT model 1 calculations. Even for a 50 datapoint dataset, a 1-sided 95% confidence interval on `mswd` extends up to 1.36. It doesn't stop at $\texttt{mswd} = 1$.

The main ideas in this section of our response are incorporated in the revision, but a discussion of model 3 calculations is beyond our remit there.

[revised manuscript text omitted]

---

## Referee Report (RR1)

Review of "Robust Isochron Calculation" Version 4 Revision by Powell et al. (gchron-2020-4-manuscript version4)

September 3, 2020

**1 Summary**

This is a greatly improved version of the manuscript. The abstract and introduction now provide a much better lead-in for the main parts of the paper. Furthermore, the authors have paid careful attention to this reviewer's specific recommendations for improving the manuscript, and for the most part have responded quite positively. The Section 1.1 On ISOPLOT and Section 1.2 Replacing ISOPLOT are very effective. The over-arching value of the paper is that: (1) introduces the use of the well-known Huber robust regression method in Geochronology research, and (2) it provides a novel algorithm, including Python code, for computing a Huber robust regression estimator where the measurements are subject to error. As such, I recommend publication of the paper.

That said, there are a few small and moderate issues and one major issue, described below, that the authors need to take care of.

**2 Small Issues**

1. The term "excess scatter" used in the Abstract and several places in the Section 1 Introduction is far too vague particularly when the focus is dropping the normality assumption for the tails of the distribution. It would be better to use language such as "when the data distribution has non-normal tails (or fat tails) associated with outliers", or "when the data distribution has higher variability in the tails than is indicated by the variability of the central part of the distribution".

2. The Abstract contains the statement: "A statistical test is provided to ensure that a central spine of the uncertainty distribution data is Gaussian." Then in the discussion connected with Table 1, it should be pointed out exactly what is the statistical test.

3. In several places the authors state in effect that a robust method produces "identical" results as least squares, this seldom literally true, and better term would be "nearly identical".

4. The author's use the term "resistant" to indicated a highly robust (e.g., with regard to breakdown point and bias control) but rather inefficient estimate. In fact "resistant" is just an alternative data-oriented term for "robust", the latter typically meaning with respect to analytic measures such as variance/efficiency, bias and mean-squared error. So the term resistant should be dropped, and in its place use "highly robust (toward outliers, or with respect to bias) but inefficient (in terms of variances) at a normal distribution".

5. The two-component normal mixture distribution was used in the seminal paper Tukey, J.W. (1960), *A survey of sampling from contaminated distributions, in: Contributions to Probability and Statistics, I. Olkin, Ed., Stanford University Press.* to show that simple trimmed means perform better than the sample mean in terms of efficiency for such distributions, including the normal distribution as a special case. This paper inspired a lot of subsequent research, e.g., by Huber and Hampel and beyond. So it wouldn't hurt to reference it.

6. What is the "Tera-Wasserburg isochron space"? Maybe all readers know what this is? Can't this be stated in a way that is for sure clear to all possible readers?

7. The sentence just below equation (2) states: "with the constant normalizing the result to be like the standard deviation for Gaussian-distributed $r$. The use of the word "like" is evidently to make this paper easier to read by a broad audience of readers. However, for basic concepts such as "an unbiased estimator", I think it is best to be precise. In this particular case, the choice of the constant 1.4826 makes `nmad(r)` an unbiased estimator of the standard deviation as the sample size goes to infinity when the data is normally distributed, and approximately unbiased in finite samples. Something like this should be put in a footnote.

8. It would be good to consistently use circumflex's to indicate estimators, e.g., $\hat{s}$ instead of $s$ in equation (2).

**3  Moderate Issues**

1. In Section 2.4 the authors state: "Efficiency at the Gaussian distribution is the ratio of the variance obtained by the optimal estimator (YORK), divided by the variance using the chosen robust estimator (in this case, SPINE).", which is quite correct. However, the subsequent sentence: "Obviously, SPINE has optimal efficiency when all r in a data set have $|r_k| < h$, when it is identical to YORK, but there is an efficiency loss associated with using SPINE for an isochron-yielding data set with any $|r_k| < h$." is quite nonsensical. Efficiency of an estimator is defined at a data distribution, not at a data set!

2. The practice of trying to define an errorchron versus an isochron based on the character of the residuals from a straight line fit seems to be a rabbit hole that one should not be caught in. After all, the robust regression could be of quite high quality (see next item) with regression residuals being quite fat-tailed and with outliers.

3. The use of mswd in the broad range of regression practice is seldom used (seems to have arisen mostly in connection with categorical data and two-way tables). But R-squared is always used as a measure of goodness of a regression fit. So why is that not used? And if it is not used on purpose, because mswd is deemed to be better, what is the justification. Something needs to be said about this.

4. The idea to somehow use `nmad(r)` as defined by equation (2) is a good one. However, using it for testing whether or not the central part of the distribution is normal, as the authors do, is not very useful. After all, the central part is usually well-modeled by a normal distribution, and it is fat non-normal tails that are the issue. One might consider tests based on the ratio of the standard deviation estimate to the `nmad(r)`, or a test based on the difference between 95% and 75% ordered data. But at the small sample sizes encountered, such a statistic may not be very useful.

**4   Main Issue**

At the end of Section 2.1 the authors state: "Note that with the sample sizes provided by most modern geochronological techniques, it is not possible to test for Gaussian behaviour, or such small departures from Gaussian behavior." For really small sample sizes, say 10 - 20 or so, formal tests such as the Kolmogorov-Smirnov test or the Anderson-Darling test are not likely to be very useful. However, normal qqplots with point-wise 95% confidence bands are very useful for detecting fat-tails and outliers.

To this I would add: I have reviewed many papers on applications of robust statistics over the years and this is the first time I have seen a paper promoting the use of robust regression without a data example showing that there is at least one potentially influential outlier (and often more than one).

NOTE: I ran least squares and a robust regression estimator on the pairs of mean values of the ellipses in Figure 7, with the result that there is essentially no difference between the LS and robust regressions, and no indication what-so-ever of outliers in the regression residuals.

So the paper needs one convincing example of a data set where there are one or more outliers that result in a clear difference between the LS and robust fits (as measured most simply by the robust standard error of the robust regression coefficients), and outliers in the regression residuals as indicated by a normal qqplot with 95% point-wise confidence intervals. Otherwise, the paper presents a solution in search of a problem.

---

## Editor Decision (ED1)

Dr. Pieter Vermeesch
University College London
+44 (0)20 3108 6369
p.vermeesch@ucl.ac.uk

5 May 2020

To: Roger Powell, University of Melbourne

Dear Professor Powell,

Thank you for submitting your manuscript entitled "Robust Isochron Calculation" to *Geochronology*. Having considered your paper, the four reviews and your response to the reviews, I have decided that your manuscript is suitable for publication in *Geochronology* after moderate to major revision. In addition to the reviewer comments, I would like to add a few thoughts of my own.

1. I agree with reviewers 3 and 4 that your paper needs to become accessible to a wider readership. I am concerned that there are only a handful of people in the geochronology community who will be able to understand the current version of your paper, or run your `Python` code. The impact of your work would greatly increase if you could explain your algorithm in plain English. Please note that I am not asking you to remove the mathematical details of the robust regression algorithm from your paper. In fact it is important that those details are retained. But much of this could be moved to the appendices.
   In your response to reviewer 4, you wrote that "Clearly it was a mistake to sweep as many of the equations as possible into the Appendix!". I disagree, and think that you could move even more content to the appendices, as long as you add enough links to them in the main text.

2. The algorithm in your paper is not the first to apply robust statistics to isochron regression. In fact, robust isochron regression is already implemented in Ken Ludwig's popular `Isoplot` add-in to `MS-Excel`. Unfortunately, the `Isoplot` user manual does not provide any details about this implementation. However, Ludwig (2003) does mention your earlier paper on the subject (Powell et al., 2002), and a personal communication from you. Does this mean that `Isoplot`'s robust regression algorithm is based on your bootstrap algorithm? I would love to read more about this in your paper. In any case, I think that you should discuss the merits and limitations of these competing algorithms in a revised version of your paper. A side-by-side comparison of the different algorithms on the same dataset would be particularly helpful.

3. In your response to reviewer 3, you wrote:

   > "the 95% confidence limit on the ages is 3.97 to 4.03 Ma with HUBER , but 3.91 to 4.09 Ma under YORK , a significant increase in reliability with HUBER."

   Precision is not the same as reliability! This brings me to an important point that was only briefly touched by the reviewers, but I think should be addressed in the revision. In your paper, you refer to isochrons that exhibit excess scatter (MSWD≫1) as 'errorchrons'. Ludwig (2003) proposes five ways to deal with these. The first is to ignore the excess dispersion; the second is to inflate the errors by a factor of $\sqrt{\text{MSWD}}$; the third is to ignore the analytical uncertainties altogether; the fourth is to quantify the dispersion as a separate free parameter; and the fifth is robust regression. Your paper only mentions options 1, 2, 3 and 5. However I would argue that the fourth

Dr. Pieter Vermeesch
University College London
+44 (0)20 3108 6369
p.vermeesch@ucl.ac.uk

option ('model-3 regression') is the best way to deal with excess scatter (Vermeesch, 2018).

The great appeal of maximum likelihood estimation is that it provides clear tests for its underlying assumptions. No such tests are available for robust regression. On the surface, the ability of your algorithm to fit errorchrons may seem like a strength. But this robustness comes at a cost, in that is no longer possible to tell the difference between 'good' and 'bad' datasets. This creates a 'garbage in, garbage out' problem. Can you explain how to deal with this?

The 'errorchron' moniker is widely used in geochronology. But I think that its negative connotation is undeserved. A high precision TIMS errorchron can have greater scientific value than a low precision LA-ICP-MS isochron. What matters is not whether the data are overdispersed or not, but rather *how much* overdispersion there is. This is exactly what model-3 regression aims to achieve. In the case of isochrons, the excess scatter can either be attributed to the initial (non-radiogenic) isotope ratios, or to diachronous isotopic closure.

The dispersion has scientific value. For example, Rioux et al. (2012) estimated the dispersion of high precision TIMS U–Pb data to estimate the residence time of zircon in a magma chamber. A robust algorithm would have completely missed this information. As technology improves, overdispersed datasets will become ever more prevalent. For example, new noble gas mass spectrometers achieve an order of magnitude improvement in $^{40}Ar/^{39}Ar$ age precision. This has revealed that even the best reference materials are not homogeneous (Heizler, 2012). So in my opinion, geochronologists might want to abandon the 'cult of MSWD' and embrace dispersion rather than banish it.

I am not asking you to provide a comprehensive overview of model-3 regression in your paper. But I do think that it is important that the revised manuscript (a) addresses the 'garbage in, garbage out' problem, and (b) warns the users that any scientifically valuable dispersion will be lost.

4. Please rewrite and extend your abstract. The late Albert Tarantola once pointed me to the following text, which I found very useful:
   `https://www.caam.rice.edu/~symes/CAAM600/abstract_scrutiny.pdf`

*Geochronology* normally gives authors four weeks to complete the revision. But I would be happy to extend this to eight weeks if you need it. Please do not hesitate to contact me if you have any questions.

Sincerely yours,

Dr. Pieter Vermeesch
Department of Earth Sciences
University College London

Dr. Pieter Vermeesch
University College London
+44 (0)20 3108 6369
p.vermeesch@ucl.ac.uk

[Figure]

*References

Heizler, M. Higher Precision: Opening A New $^{40}$Ar/$^{39}$Ar Can Of Worms. In *AGU Fall Meeting Abstracts*, 2012.

Ludwig, K. R. Mathematical–statistical treatment of data and errors for $^{230}$Th/U geochronology. *Reviews in Mineralogy and Geochemistry*, 52(1):631–656, 2003.

Powell, R., Hergt, J., and Woodhead, J. Improving isochron calculations with robust statistics and the bootstrap. *Chemical Geology*, 185(3-4):191–204, 2002.

Rioux, M., Lissenberg, C. J., McLean, N. M., Bowring, S. A., MacLeod, C. J., Hellebrand, E., and Shimizu, N. Protracted timescales of lower crustal growth at the fast-spreading East Pacific Rise. *Nature Geoscience*, 5(4):275–278, 2012.

Vermeesch, P. `IsoplotR`: a free and open toolbox for geochronology. *Geoscience Frontiers*, 9:1479–1493, 2018.

---

## Author Response (AR2)

**response to associate editor**

We hope that your first point—concerning considering a dataset with non-Gaussian residuals—is covered appropriately under the "Main Issue" in the response to the additional review by reviewer 4 below. As stated there, the point of using SPINE is not to defend against serious outliers but to extend the range of reliable age determination beyond that possible with YORK

5     In fact, we have performed model 3 calculations on sample 0708, and also on some simulated datasets. The results are interesting, but they make it seem unwise for us to include model 3 calculations in Appendix A. Our first concern is that we don't think it is appropriate to consider the scatter that is to be handled by geological variation to start at $\texttt{mswd} = 1$. In the case of sample 0708, data scatter is consistent with analytical uncertainties for $\texttt{mswd} < 1.2$, at 95% confidence. Doesn't using $\texttt{mswd} = 1$ to flag the necessity of including geological variation potentially overestimate the variation?

10     The second concern does not show for sample 0708 because the distribution of the residuals is close to Gaussian, although far enough away from Gaussian that $\texttt{mswd}$ "fails". One consequence is that YORK and SPINE give very similar ages. But, in general, it seems that the age determined with model 3 is insensitive to the geological variation, however the age is determined. Thus, if the data structure means that YORK has functioned sub-optimally because of outliers, a model 3 calculation performed using YORK (as done in isoplot) may give a rather different result to the same calculation done using SPINE. Such outliers may

15 well be a valid part of the geological variation, so they are not in fact outliers in the context of model 3. Generally, then, won't using YORK for model 3 calculations potentially result in overestimated geological variation, as well as an unreliable age?

    The changes to the manuscript are indicated in the response to the additional review, and are in blue in the revised version below.

**response to additional review of reviewer 4**

20 The additional comments by reviewer 4 are much appreciated. Before responding to these, a left-over matter from this reviewer's first review, is the observation that the iteration used to find $\theta$—the parameters of the best-fit line—is likely to be unreliable in general. We asserted that attempts to find an iteratively reweighted least squares iteration had failed, and indeed that all the simulations undertaken had been successful with the iteration given. Nevertheless, following submission of the revision, the first author tried again, and has indeed derived an iteratively reweighted least squares iteration (based on Maronna

25 *et al.*, 2019, Section 4.5.2, but including the analytical uncertainties). This is now adopted in the python code in Appendix C as a more reliable (and also more concise) approach. The logic of the original iteration is still retained in Appendix B as it is necessary for the derivation of the covariance matrix of $\theta$.

**Small issues**:

1. While we hope to retain the term "excess scatter" as a short-hand way of referring to the consequence of fat tails, a form
30     of words based on the reviewer's commentary is now given in a footnote (following the suggestion of using footnotes by the reviewer in his point 7), to clarify this usage.

2. It is quite correct that the spine width, $s$ (or, strictly, $\hat{s}$) says nothing about whether the centre of the distribution of $r$ is Gaussian or not. The wording related to this has been tightened up.

3. If all the $|r_k| \leq h$ in a dataset, then SPINEresults are literally identical to YORK and $\sum \rho(r_k)$ does equal $(n-2)\texttt{mswd}$
35     then. For example, for $n = 10$, some 17% of strictly Gaussian-distributed data have datasets with no $|r_k| > h$. Otherwise the reviewer is correct that SPINE results are only nearly identical to YORK results for "good" datasets. This has been clarified in the text.

4. In using the word "resistant", we were following Huber & Rochetti (2009, Sect 1.2.3), though now we see that Maronna *et al.* (2019) does not use the resistant–robust distinction. Resistant is no longer used in the manuscript.

40 5. Good idea: Tukey reference added.

6. The term "Tera-Wasserburg" has now been deleted. It is unnecessary here—it just refers to this particular type of isotope plot.

7. A footnote has been added relating to the `nmad` normalisation.

8. While we understood that using circumflex on estimators is strictly correct, we are hoping to forgo that for the sake of keeping the notation less cluttered, and for the main audience of our paper.

**Moderate issues**:

1. Some confusion in the mind of the first author regarding efficiency has now been dispelled. The words relating to efficiency should now be appropriate.

2. Agreed, robust data fitting methods are good, and can be applied to datasets that are more scattered than those usually encountered in geochronology. The problem is that in geochronology there is an expectation of a statistical test that suggests whether a dataset has age significance (isochron) or not (errorchron). For example, in the covering letter from the associate editor requesting us to submit a revision, in which he asked how do we stop the "rubbish in–rubbish out" problem in data fitting (which we took to mean an errant dataset giving an age that is geologically meaningless). So we do make a distinction—akin to the one based on `mswd`—via the information in Table 1. This is a rabbit hole—but unfortunately one that we have been forced down. We accept that the good behaviour of SPINE makes such a decision unnecessary. But the `mswd` limit was less to prevent the potentially-errant behaviour of YORK (mostly not acknowledged, or even ignored), and more to do with assessing whether a dataset gives a geological meaningful age (deemed to be when the data were consistent with strictly Gaussian-distributed data).

3. The use of `mswd`, rather than $R^2$, seems to be a historical accident in geochronology going back at least to the early York and other papers in the mid-60s. We don't know why $R^2$ was never adopted. Certainly $R^2$ is not given by ISOPLOT.

4. Yes. Whereas it is currently asserted in the manuscript that $nmad(\mathbf{r})$ somehow discerns whether the central part of the distribution of $\mathbf{r}$ is effectively Gaussian, in fact this is plainly not correct. The spine width cannot do that. In the context that it is used, all that is required of the spine width is that it reflects the presence of a spine in the data. Nothing more. The wording in the manuscript has been modified to reflect this. It is not necessary to show that the central part of the distribution of the residuals is Gaussian for SPINE to work, and, as the reviewer points out, the central part of the distribution will tend to be approximately Gaussian anyway.

**Main issue**:

The usual demonstration of the deleterious effect of outliers on least squares (equivalently, YORK) uses datasets with very obvious outliers, e.g. Maronna *et al.* (2019), Fig. 4.3. Particularly in geochronology, such datasets would not be used in data-fitting for age determination. If used, they may well "cleaned up" first (but with the problems that can arise from that, as discussed in the robust statistics literature). Nevertheless, as expected, SPINE does work well in the presence of egregious outliers, finding the main trend of the data.

The purpose of using SPINE is not handling obvious outliers, rather the purpose is to extend the range of datasets for which a reliable age can be determined, where the deleterious effect of data scatter is real, but not dramatic as it is in the Maronna example. This effect can be seen clearly in new simulations, using for example residuals distributed as 10%10N with $n = 10$. For 10,000 simulated datasets, with a nominal age of 4 Ma, for the datasets that "fail" `mswd`, a 95% confidence interval on the SPINE age is $4\pm0.033$ Ma, whereas for YORK it is $4\pm0.092$ Ma. Also, the age difference between the two methods, normalised by the $\sigma$ on the SPINE age, is $\pm5.54$. Such calculations showing the reliability of SPINE, beyond the range of data scatter that YORK can handle, are now summarised in Section 2.4.

It is not clear that what the reviewer wants to see in a graphical demonstration of the success of SPINE is actually easy to achieve with a geochronologically-available or -sensible dataset. Instead, we show this via a simulated dataset in Appendix E in the revision, also using a quantile-quantile plot, hoping that they might be used more widely in this context.

[revised manuscript text omitted]

---

## Author Response (AR3)

**response to associate editor**

Hopefully the new abstract meets with your satisfaction. The only thing that has changed in the manuscript is the abstract

There is much to discuss concerning model 3! Maybe we can continue by email, separately from the matter of the current paper? I (the first author) would want to do more playing with simulations before it would be worth doing that?

**Robust Isochron Calculation**

Roger Powell[1], Eleanor CR Green[1], Estephany Marillo Sialer[1], and Jon Woodhead[1]

[1]School Earth Sciences, The University of Melbourne, Vic 3010, Australia

**Correspondence:** Roger Powell (powell@unimelb.edu.au)

5 **Abstract.**

The standard classical statistics approach to isochron calculation assumes that the distribution of uncertainties on the data arising from isotopic analysis is strictly Gaussian. This effectively excludes from consideration datasets that have more scatter, even though many appear to have age significance. A new approach to isochron calculations is developed in order to circumvent this problem requiring only that the central part of the uncertainty distribution of the data defines a "spine" in the trend of the

10 data. This central spine can be Gaussian but this is not a requirement. This approach significantly increases the range of datasets from which age information can be extracted but also provides seamless integration with well-behaved datasets, and thus all legacy age determinations. The approach is built on the robust statistics of Huber (1981), but using the data uncertainties for the scale of data scatter around the spine, rather than a scale derived from the scatter itself, ignoring the data uncertainties. This robust data-fitting reliably determines the position of the spine when applied to data with outliers, but converges on the classical

15 statistics approach for datasets without outliers. The spine width is determined by a robust measure, the normalised median absolute deviation of the distances of the data points to the centre of the spine, divided by the uncertainties on the distances. A test is provided to ascertain that there is a spine in the data, requiring that the spine width is consistent with the uncertainties expected for Gaussian-distributed data. An iteratively-reweighted least squares algorithm is presented to calculate the position of the robust line and its uncertainty, accompanied by an implementation in Python.